# 5mC modification orchestrates choriogenesis and fertilization by preventing prolonged *ftz-f1* expression

Zheng Zhao[1,2,3,5], Liang Li[1,5], Ruichen Zeng[1], Liangguan Lin[1], Dongwei Yuan[1], Yejie Wen[1], Na Li[1,3], Yingying Cui[1], Shiming Zhu[1], Zhi-Min Zhang[4], Sheng Li [1,2,3] ✉ & Chonghua Ren [1,3] ✉

DNA methylation at the fifth position of cytosine (5-methylcytosine, 5mC) is a crucial epigenetic modification for regulating gene expression, but little is known about how it regulates gene expression in insects. Here, we pursue the detailed molecular mechanism by which DNMT1-mediated 5mC maintenance regulates female reproduction in the German cockroach, *Blattella germanica*. Our results show that *Dnmt1* knockdown decreases the level of 5mC in the ovary, upregulating numerous genes during choriogenesis, especially the transcription factor *ftz-f1*. The hypomethylation at the *ftz-f1* promoter region increases and prolongs *ftz-f1* expression in ovarian follicle cells during choriogenesis, which consequently causes aberrantly high levels of 20-hydroxyecdysone and excessively upregulates the extracellular matrix remodeling gene *Mmp1*. These changes further impair choriogenesis and disrupt fertilization by causing anoikis of the follicle cells, a shortage of chorion proteins, and malformation of the sponge-like bodies. This study significantly advances our understanding of how DNA 5mC modification regulates female reproduction in insects.

Epigenetic DNA modification provides an additional layer of genome regulation, enhancing the developmental plasticity of organisms. DNA methylation at the fifth position of cytosine (5-methylcytosine, 5mC) is the most common and abundant DNA modification in animals[1]. This particular modification is initially established by de novo DNA methyltransferase 3 homologs (DNMT3) and subsequently maintained by DNA methyltransferase 1 homologs (DNMT1) in mammals[2,3]. By orchestrating gene expression, 5mC modification assumes a pivotal role in governing fundamental processes, such as genomic imprinting, X chromosome inactivation, and the alteration of chromosome stability and chromatin structure[4,5]. In most orders of insects, 5mC modification is widespread at various levels[6] and contributes to caste

differentiation, longevity, reproduction, and embryogenesis[7–9]. However, there is limited knowledge regarding the mechanistic links between 5mC modification and insect development, particularly in terms of transcriptional regulation[10]. Therefore, we are currently at the dawn of a golden epigenomic age for research on DNA methylation in insects[11].

Insects, the most diverse animal group on Earth, owe part of their success to their high reproductive capacity. Female insect reproduction generally involves the production of yolky oocytes, mating, and egg-laying. The ovaries play a crucial role in these processes, with distinct phases, including previtellogenesis, the accumulation of vitellogenin (Vg) proteins during vitellogenesis, and the deposition of

[1]Guangdong Provincial Key Laboratory of Insect Developmental Biology and Applied Technology, Guangzhou Key Laboratory of Insect Development Regulation and Application Research, Institute of Insect Science and Technology & School of Life Sciences, South China Normal University, Guangzhou 510631, China. [2]Guangdong Laboratory for Lingnan Modern Agriculture, Guangzhou 510631, China. [3]Guangmeiyuan R&D Center, Guangdong Provincial Key Laboratory of Insect Developmental Biology and Applied Technology, South China Normal University, Meizhou 514779, China. [4]College of Pharmacy, Jinan University, 510632 Guangzhou, China. [5]These authors contributed equally: Zheng Zhao, Liang Li. ✉e-mail: lisheng@scnu.edu.cn; renchonghua111@m.scnu.edu.cn

the eggshell or chorion during choriogenesis[12]. During pre-vitellogenesis, follicle cell growth, driven by insulin signaling and the target of rapamycin[13]. Vitellogenesis is mainly controlled by juvenile hormone (JH) and/or 20-hydroxyecdysone (20E, the active form of insect steroid hormone converted from ecdysone) in different insect orders[13–15]. In most, if not all, insects, 20E regulates the final oogenesis stage, choriogenesis[16–18]. Notably, the ovaries become the primary source of ecdysteroids in adult females of many species, as the larval/nymphal-specific ecdysteroid synthesis (steroidogenesis) tissue, the prothoracic gland, degenerates post metamorphosis[19]. However, the regulatory mechanism of ovarian steroidogenesis is elusive. The 20E level in the ovaries of the German cockroach gradually increases and peaks before oviposition, and ovarian 20E triggers choriogenesis in an autocrine fashion[19]. In the silkworm, *Bombyx mori*, topical application of 20E during choriogenesis negatively impacts follicle cell development, and a particularly high level of 20E is able to induce follicle cell death[12].

5mC modification has been reported to play a role in female reproduction in various insect species, such as the German cockroach, *B. germanica*; the silkworm, *B. mori*; the parasitic wasp, *Nasonia vitripennis*; the brown plant hopper, *Nilaparvata lugens*; and the milkweed bug, *Oncopeltus fasciatus*[7,9,20–23]. However, the underlying regulatory mechanism remains largely unknown. Compared to those in other insect orders, 5mC levels are higher in cockroaches (e.g., the German cockroach)[6], which are representative and classical hemimetabolous insects and suitable for studying panoistic ovary maturation and oogenesis. Previously, female reproduction and its regulation have been well studied in the German cockroach[18,24–28]. In this study, using the German cockroach as a model insect, we elucidated how 5mC modification regulates female reproduction through controlling *ftz-f1* medicated choriogenesis and fertilization.

## Results

### DNMT1-mediated 5mC maintenance is indispensable for choriogenesis and fertilization

To investigate the relationship between 5mC modification and female reproduction in *B. germanica*, we first examined the spatiotemporal pattern of 5mC modification in female adults. The spatial pattern of 5mC was detected in six tissues/organs, and higher levels in the ovaries, thorax, and integuments compared to the heads, legs, and midguts (Fig. 1a). Subsequently, the temporal pattern of 5mC in the ovaries during the whole first reproductive period (8 days) was evaluated. A decreasing trend was observed from day 1 to day 4 post-adult emergence (PAE), followed by an increase until oviposition (Fig. 1a'). Focusing on the DNMT1 protein, which has been shown to be essential for embryogenesis in this species[9], the functional domains are highly conserved throughout evolution (Supplementary Fig. 1a). The full-length DNMT1 of *B. germanica* shares 45% amino acid sequence identity with human DNMT1. Furthermore, the expression level of *Dnmt1* in the ovaries during the first gonadal cycle was evaluated by using quantitative real-time PCR (qRT-PCR), and its developmental pattern was consistent with the genomic 5mC tendency (Fig. 1b). Together, these spatiotemporal patterns suggest that the DNMT1-induced 5mC maintenance in ovaries may play a crucial role in female reproduction.

To test the hypothesis stated above, we performed RNA interference (RNAi) experiments in female adults. dsRNA was designed to target the *Dnmt1*, and the gene expression was significantly reduced upon ds*Dnmt1* treatment (Supplementary Fig. 1b). Dot blot analyses revealed that, compared to the negative control (ds*CK*) group, the 5mC levels were significantly reduced in the ds*Dnmt1* group (~40%) (Fig. 1c). To verify these results, a much more sensitive antibody-independent LC–MS/MS analysis was performed, and almost identical results were obtained (Fig. 1d). The impact of RNAi-mediated decreases in 5mC levels on female reproduction in *B. germanica* was then examined.

In the ds*Dnmt1* group, all oothecae were atrophied and wizened, and all the eggs inside the oothecae failed to hatch (Fig. 1e). Interestingly, neither mating nor ootheca-laying rates were affected (Supplementary Fig. 1c, d). We next examined the possible defects of ds*Dnmt1* treatment on eggs and ovaries. Firstly, DAPI staining of eggs was carried out to examine embryogenesis after ootheca-laying. Abundant energids in the yolk mass were observed in the ds*CK*-treated eggs, which tended to concentrate at the ventral surface at 60 h and developed into early embryos at 84 h[29,30]. Conversely, the ds*Dnmt1*-treated eggs did not show cleavage energids in the yolk mass, only faint and diffuse staining was observed, and embryogenesis did not occur. The ds*Dnmt1*-induced phenotypic defects were similar to those in the unmated negative control group, as German cockroaches do not undergo parthenogenesis (Fig. 1f). Moreover, pronuclear staining of eggs that had just entered the oothecae revealed both female pronuclei from egg and male pronuclei from sperm in the ds*CK*-treated eggs, while no male pronuclei were detected in ds*Dnmt1*-treated or unmated eggs (Fig. 1g). Importantly, the sponge-like body, an atretic structure appearing at the anterior pole of the basal oocyte through which sperms enter maturing egg for fertilization[31], was severely malformed in the ds*Dnmt1*-treated insects (Fig. 1h). Together, the experimental data demonstrate that ds*Dnmt1*-induced malformation of sponge-like body disrupts fertilization and further terminates embryogenesis.

We further investigated other phenotypic defects in the ds*Dnmt1*-treated ovaries. There was no difference in weight or vitellogenin incorporation into oocytes between the ds*Dnmt1* and ds*CK* groups (Supplementary Fig. 1e, f). The ovarian follicle cells did not exhibit any noticeable effects on day 7 PAE (during the vitellogenesis stage) after ds*Dnmt1* treatment. However, on day 9 PAE (during the choriogenesis stage), the follicle cell nuclei in the ds*Dnmt1*-treated ovaries displayed irregular and condensed shapes, and these follicle cells displayed signs of cytoskeletal disorganization (Fig. 1i). Additionally, the tunica propria, which covers the follicle cells[31], exhibited a reticular distribution of hollows in the ds*Dnmt1* group (Supplementary Fig. 1g). These findings suggest that DNMT1-mediated 5mC maintenance is essential for timely choriogenesis and proper fertilization in the German cockroach.

### Hypomethylation-induced excessive 20E impairs choriogenesis and fertilization

Given the distinct phenomena induced by ds*Dnmt1* on day 7 PAE and day 9 PAE, RNA-Seq was performed on ovaries at both time points. In the ds*CK* group, there was a significant difference in the transcriptomes between day 7 and day 9, as revealed by principal component analysis (PCA). However, the transcriptomes of the ds*Dnmt1* group on both day 7 and day 9 were closely resembled that of the ds*CK* group on day 7, suggesting that ds*Dnmt1* treatment did not have a significant effect on day 7 but delayed developmental progression on day 9 (Fig. 2a). Volcano plot analysis supported this finding, revealing approximately 300 differentially expressed genes (DEGs) between the ds*Dnmt1* and ds*CK* groups on day 7, and around 3000 DEGs on day 9. Notably, the number of upregulated genes exceeded the down-regulated genes (1868/1306) on day 9 (Fig. 2b). These transcriptomic changes were consistent with the morphological changes observed at both time points, with insignificant effects on day 7 but significant effects on day 9 (Fig. 1i).

To investigate the effect of ds*Dnmt1* on choriogenesis, we analyzed the upregulated DEGs on day 9 PAE and examined their enrichment in KEGG pathways. Our analysis revealed that several nutrition-related pathways (e.g., cAMP signaling, PI3k-Akt signaling, insulin signaling, AMPK signaling, insulin secretion, biosynthesis of amino acids) and the insect hormone biosynthesis pathway were activated and enriched (Supplementary Fig. 2a), implying that nutrition or/and hormone might be involved in the DNMT1-regulated choriogenesis. To test this hypothesis, exogenous

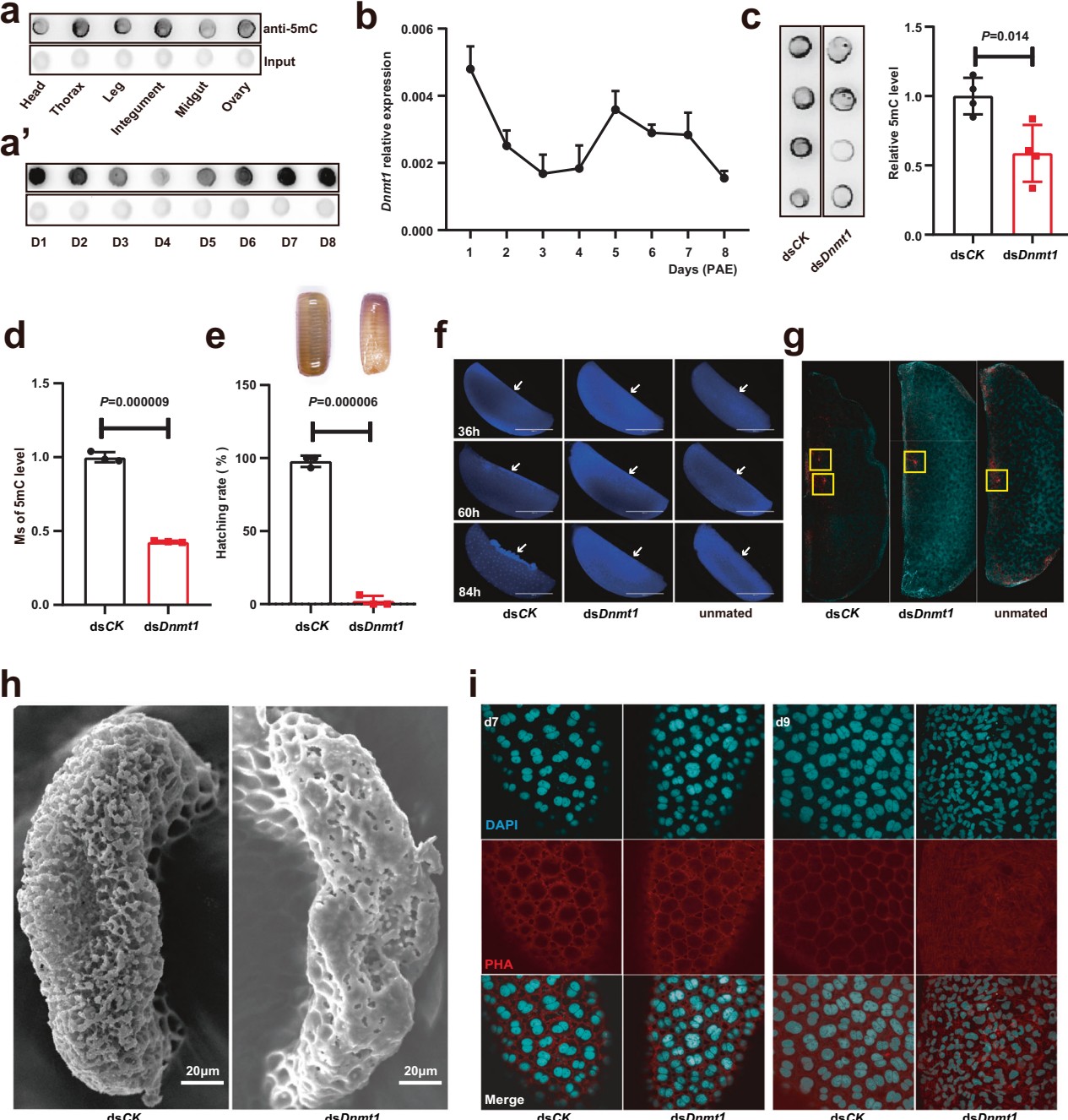

**Fig. 1 | DNMT1-mediated 5mC maintenance is indispensable for choriogenesis and fertilization. a, a′** Spatial 5mC modification was detected in six different tissues, and temporal 5mC levels in ovaries during the first reproductive period were analyzed by dot blotting using a 5mC-specific antibody. The same amounts of gDNA were loaded as input in each panel. **b** Gene expression patterns of *Dnmt1* in the ovaries during the first reproductive period were detected by qRT-PCR, *n* = 3 biologically independent samples. **c** 5mC levels were detected after RNAi of *Dnmt1* by dot blotting, and the density for each sample was calculated, *n* = 4 biologically independent samples. **d** LC–MS/MS detection for the 5mC levels under dsRNA treatment, *n* = 3 biologically independent samples. **e** Calculation of hatching rates under RNAi of *Dnmt1* and negative control. The atrophied and wizened phenotypes of oothecae in ds*Dnmt1* group compared with ds*CK* control group were showed, *n* = 3 biologically independent experiments. **f** DAPI staining was used to observe the development of embryos or unfertilized eggs, 36 h, 60 h, and 84 h post ootheca-laying were selected. The arrows point to the position of embryogenesis. More than three eggs were observed in each treatment group. **g** Pronucleus staining for the eggs using α-tubulin antibody. More than five eggs were observed in each treatment group. **h** SEM observation of sponge-like bodies on eggs, 9 of 13 sponge-like bodies were observed abnormal. **i** DAPI and PHA staining for nuclei and actin protein in ovarian follicle cells. Not less than three eggs were observed in each group. Of note, one more day is added and there are 9 days in the first reproductive period under dsRNA injection condition. Data are mean ± sd, the differences were analyzed by two-tailed Student's *t* test. Source data are provided as a Source Data file.

glucose, amino acids (Aa) mixture, and insulin were injected into female adults, however, no significant decrease in embryonic development or hatching rate was observed for any of the treatments (Supplementary Fig. 2b, c). Thus, we focused on the insect hormone biosynthesis pathway, since *phantom* (*phm*) and *shadow* (*sad*), both of which are involved in steroidogenesis, were identified upregulated in ds*Dnmt1* group (Fig. 2b, c). As aforementioned, ovary is the main source of steroidogenesis and circulating 20E in

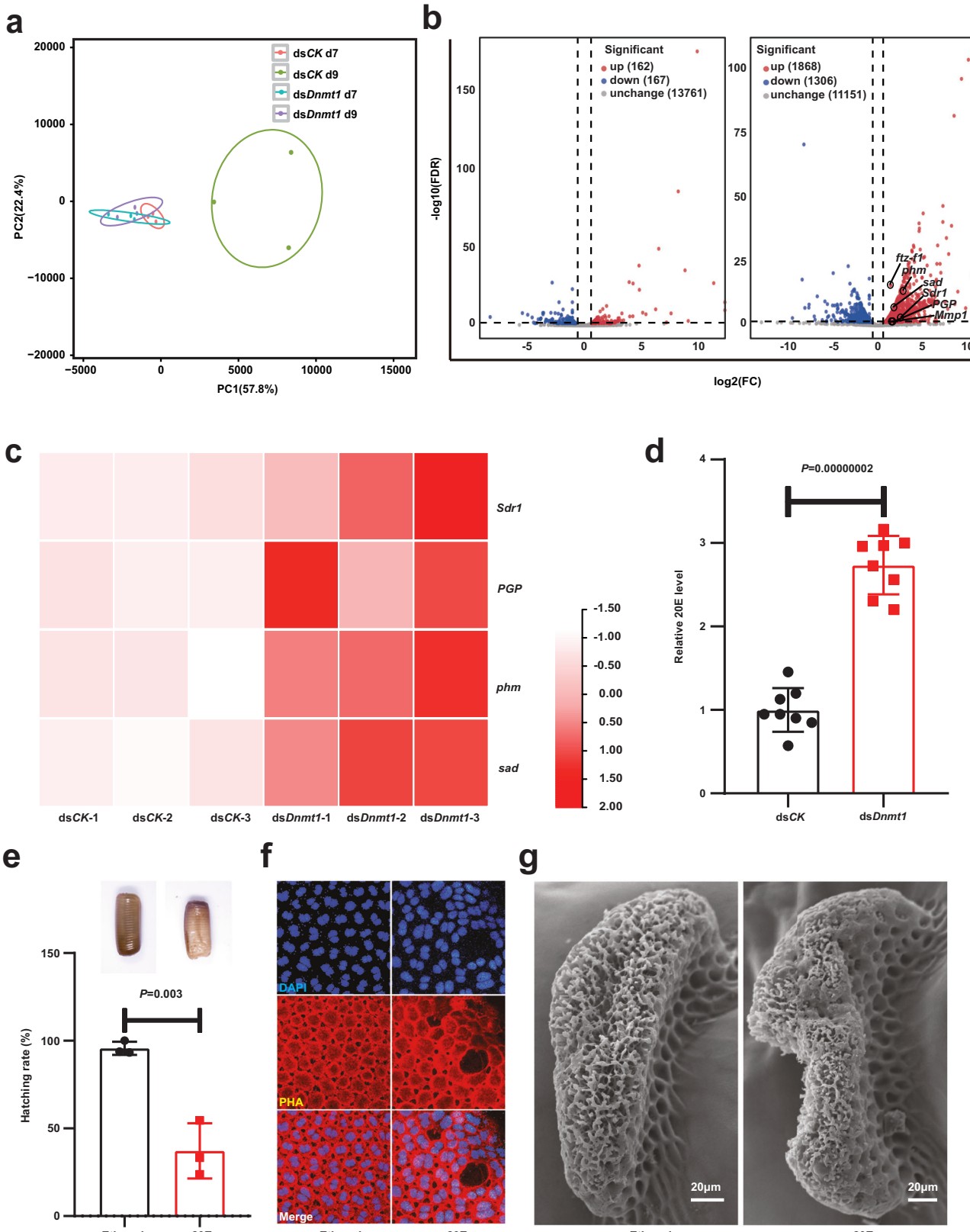

female adults[11,19]. Notably, we observed a 3-fold increase in the level of 20E in the ovary of the dsDnmt1 group on day 9 PAE (Fig. 2d). To confirm the effect of 20E on choriogenesis, we injected a low dose of 20E into female adults. Similar to previous reports in *B. mori*[12], most 20E-treated oothecae atrophied and wizened, and the majority of the eggs inside them failed to hatch (Fig. 2e). Moreover, the nuclei and cytoskeletal structures of some follicle cells became abnormally shaped (Fig. 2f), and most of the sponge-like bodies in the 20E-treated animals were malformed (Fig. 2g). In short, the dsDnmt1-induced phenotypic defects in choriogenesis and fertilization should be attributed in part to the aberrant elevation of 20E levels during choriogenesis.

**Fig. 2 | Hypomethylation-induced excessive 20E impairs choriogenesis and fertilization. a** PCA of gene expression among four groups: the ds*CK* day 7 (d7), the ds*CK* day 9 (d9), the ds*Dnmt1* d7, and the ds*Dnmt1* d9. **b** Volcano plot analysis of the DEGs under ds*Dnmt1* treatment conditions on d7 and d9. Green dots indicate downregulated genes, and red dots indicate upregulated genes. **c** Heat map of genes enriched in the pathways of insect hormone biosynthesis. **d** 20E levels detection under ds*Dnmt1* treatment on day 9, $n = 8$ biologically independent samples. **e** Calculation of hatching rates under exogenous 20E treatment. The atrophied and wizened phenotypes of oothecae in 20E group compared with ethanol control group were showed, $n = 3$ biologically independent experiments. **f** DAPI and PHA staining for nuclei and actin protein in ovarian follicle cells. **g** SEM observation of sponge-like bodies on eggs under exogenous 20E treatment. 8 of 10 sponge-like bodies were observed abnormal. Data are mean ± sd, the differences were analyzed by two-tailed Student's *t* test. Source data are provided as a Source Data file.

## Hypomethylation-increased and -prolonged *ftz-f1* expression causes excessive steroidogenesis

To unravel the connection between differential gene expression and 5mC modification, we performed whole-genome bisulfite sequencing (WGBS) to analyze differences in 5mC levels in ovaries between the ds*Dnmt1* and ds*CK* groups on day 9 PAE (Supplementary Fig. 3a). In accordance with previous studies in *B. germanica*, methylation modifications were found predominantly at cytosines within the CpG dinucleotide[9], while no noticeable cytosine methylation at CHG or CHH sites (Fig. 3a, b). Importantly, whole-genomic methylation levels at CpG sites decreased from 15% to 6% ($P = 0.001$) in the ovaries of ds*Dnmt1*-treated insects (Fig. 3a). Significant differences in DNA methylation at CpG sites between the ds*Dnmt1* and ds*CK* groups were also revealed by PCA (Supplementary Fig. 3c). Compared to the ds*CK* group, 99.97% differentially methylated regions (DMRs) were observed as hypomethylated (hypo-) DMRs (205842) in the ds*Dnmt1* group, and only 0.03% hypermethylated (hyper-) DMRs (57) were found (Fig. 3b), showing that most DMRs were produced by ds*Dnmt1*-induced hypomethylation. Moreover, in the ds*Dnmt1* group, significant decreases in 5mC levels were observed in all gene regions of the genome, including the Up 2k region, the 5′-untranslated region (5′ UTR), the exons, the introns, the 3′-untranslated region (3′UTR), and the Down 2k region (Supplementary Fig. 3d).

Then, we aimed to establish the relationship between alterations in steroidogenesis and changes in 5mC levels in the ovaries of ds*Dnmt1*-treated insects on day 9 PAE. We first examined DEGs directly involved in steroidogenesis in the ds*Dnmt1* group. Although *phm* and *sad* were upregulated, no hypomethylation was observed in the Up 2k or promoter region of these genes (Fig. 3c), suggesting that the upregulation of *phm* and *sad* was not directly caused by 5mC level changes in their promoter regions. Then, genes involved in regulating steroidogenesis and 20E signaling, such as *ecdysone receptor* (*EcR*)[32], *Ecdysone-induced protein 75* (*E75*)[33], *Ecdysone-induced protein 93* (*E93*)[34], and *fushi tarazu factor 1* (*ftz-f1*)[35] were evaluated in the ds*Dnmt1* group. Hypomethylation occurred in the promoter and Up 2k regions of *EcR*, while *EcR* was only slightly upregulated (1.32-fold, $P = 0.051$); no significant differences were observed for these parameters of *E75*; *E93* expression was slightly upregulated but no hypomethylation was found in its promoter or Up 2k region; notably, *ftz-f1* was significantly upregulated and hypomethylation significantly occurred in its Up 2k and promoter regions (Fig. 3c). The composite analyses suggest that DNMT1 may inhibit steroidogenesis by maintaining 5mC levels in the promoter regions of *ftz-f1*.

Next, a correlation analysis was performed to determine the relationship between gene expression changes and 5mC decreases in the promoter region between the ds*Dnmt1* and ds*CK* groups. A stronger correlation was observed between gene upregulation and hypomethylation in the promoter regions than that of gene downregulation, and the number of upregulated genes was 1.5-fold higher than that of downregulated genes (413/275) (Fig. 3d). These data indicate that 5mC modification generally plays a more prominent role in repressing gene expression in the ovaries. Subsequently, we analyzed the 413 hypomethylated and upregulated genes in top-left quadrant, eight transcription factor (TF) genes, *huckebein* (*hkb*), *ftz-f1*, *Mi-2*, *Hormone receptor-like in 39* (*Hr39*), *bric a brac 2* (*bab2*), *Cyclic-*

*AMP response element binding protein A* (*CrebA*), *nejire* (*nej*), and *deformed wings* (*dwg*), which might regulate the expression of steroidogenesis genes were screened out (Fig. 3e). Next, the top six TF genes (>2-fold upregulation) were knocked down individually (Supplementary Fig. 3e), and only ds*ftz-f1* treatment significantly decreased the expression of *phm* (Fig. 3f). As confirmed by qRT-PCR analysis, *Dnmt1* RNAi caused approximately 4-fold upregulation of *ftz-f1* in the ovaries on day 9 PAE (Supplementary Fig. 3f). Importantly, knocking down *ftz-f1* also resulted in a significant decrease of *EcR* expression (Supplementary Fig. 3g); and moreover, knocking down of *phm*, *ftz-f1*, *EcR*, or *taiman* (*tai*, the co-activator of *EcR*) significantly reduced 20E levels (Fig. 3g and Supplementary Fig. 3h–S3j). These findings suggest that FTZ-F1 might not only act as a transcription factor promoting steroidogenesis but also as a competence factor working with Tai to enhance EcR activity for 20E signaling[36,37]. Altogether, we conclude that ds*Dnmt1*-caused hypomethylation increases and prolongs *ftz-f1* expression, leading to excessive steroidogenesis.

## *ftz-f1* promoter undergoes 5mC modification for preventing prolonged *ftz-f1* expression

We next focused on the potential link between ds*Dnmt1*-induced hypomethylation and prolonged *ftz-f1* expression on day 9 PAE. To validate the reduction in 5mC levels in the *ftz-f1* promoter region, the comprehensive 5mC levels around the *ftz-f1* in the genome was checked using WGBS data. As expected, a sharp decrease of 5mC level in the *ftz-f1* promoter region was observed in the ds*Dnmt1* group (Fig. 4a). Moreover, the high-throughput next-generation sequencing-based bisulfite sequencing PCR (NGS-BSP) was employed, and sharp reductions in 5mC levels were observed in six amplicons of the *ftz-f1* promoter region (~7 kb) (Fig. 4b). We also evaluated the temporal pattern of 5mC levels in the promoter of *ftz-f1* during the first gonadal cycle using NGS-BSP. The same 6 amplicons were analyzed, and there was an evident decreasing trend in amplicons 3 and 5 from day 1–3 PAE to day 4-6 PAE, followed by an increase until oviposition (Fig. 4c). Importantly, under our rearing conditions, *ftz-f1* expression remained low until day 6 PAE and reached its peak at the end of vitellogenesis (day 7), and then declined during choriogenesis (Fig. 4d). In general, the 5mC levels in these two promoter regions of *ftz-f1* are negatively correlated with *ftz-f1* expression levels, indicating that the *ftz-f1* promoter undergoes DNMT1-mediated 5mC modification to prevent prolonged *ftz-f1* expression.

## ds*Dnmt1*-induced defects are partially rescued by *ftz-f1* knockdown

To confirm the negative regulatory effect of *Dnmt1* on *ftz-f1* expression, ds*Dnmt1* treatment and ds*ftz-f1* treatment were conducted simultaneously. Co-treatment with ds*ftz-f1* attenuated the ds*Dnmt1*-induced upregulation of *ftz-f1* (Fig. 5a). Notably, the excessive 20E levels induced by ds*Dnmt1* were also partially attenuated by the co-treatment of ds*ftz-f1* (Fig. 5b). Since ds*ftz-f1* treatment can completely disrupt oviposition process, the co-treatment with ds*Dnmt1* increased the oviposition rate back to a relatively high level (~80%) (Fig. 5c). Additionally, we paid more attentions on the obvious phenotypic defects on follicle cells and sponge-like bodies. The co-treatment with ds*ftz-f1* partially rescued the ds*Dnmt1*-

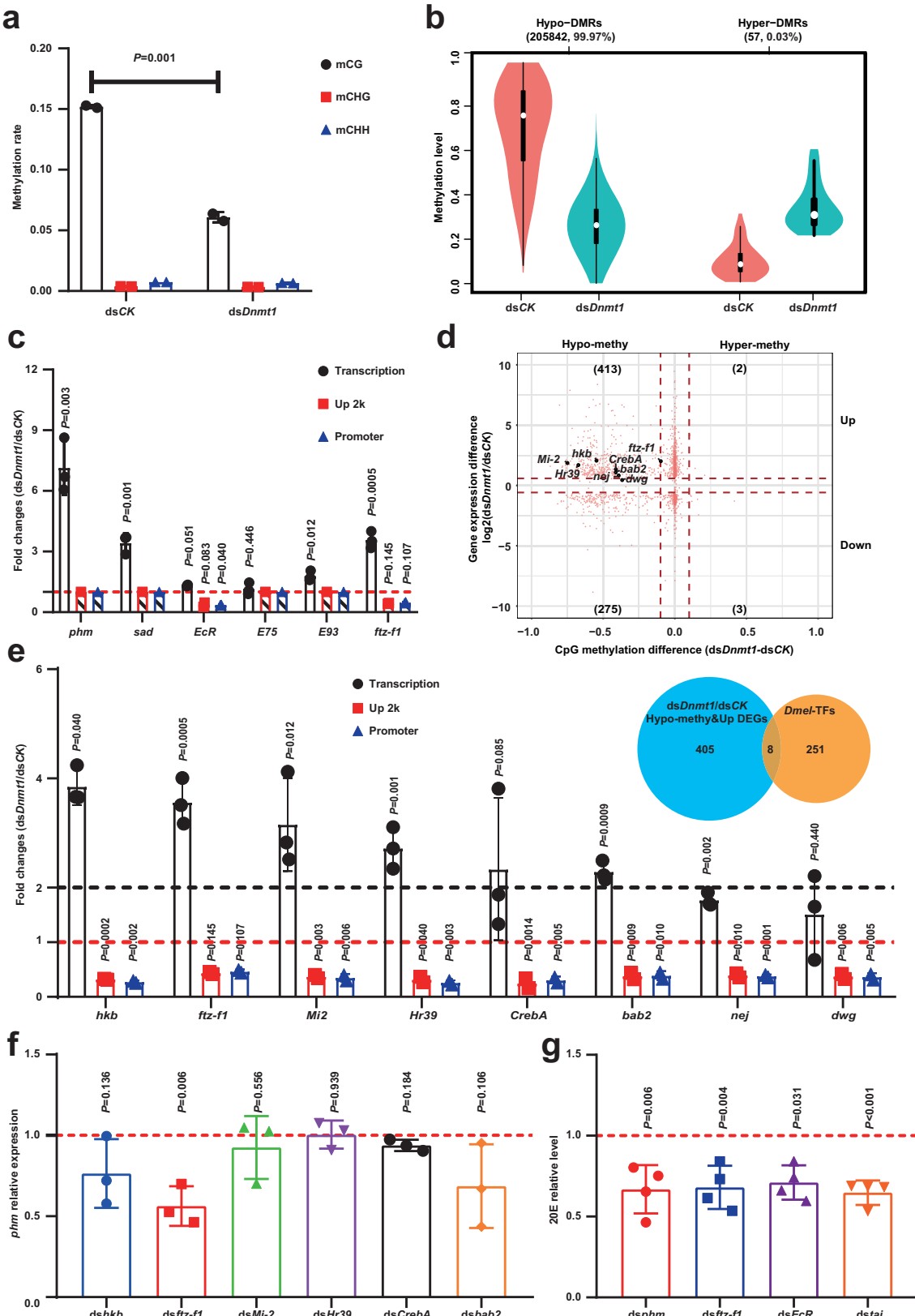

induced follicle cell disorganization and sponge-like body malformation (Fig. 5d, e). The results of co-injecting ds*Dnmt1* and ds*ftz-f1* further demonstrate that DNMT1-maintained 5mC levels in the *ftz-f1* promoter regions inhibit *ftz-f1* expression, and the ds*Dnmt1*-increased and -prolonged *ftz-f1* expression disrupt choriogenesis and fertilization.

## 5mC modification prevents *ftz-f1*-induced *Mmp1* expression and anoikis

To investigate how hypomethylation-induced *ftz-f1* upregulation regulates choriogenesis and fertilization, we conducted additional investigations since exogenous 20E treatment only partially phenocopy ds*Dnmt1*-induced defects. Focusing on the abnormal disorganization

**Fig. 3 | Hypomethylation-induced *ftz-f1* overexpression regulates steroidogenesis in follicle cells. a** Calculated cytosine methylation rates for CG, CHG, and CHH motifs, where H equals A, T, or C. Methylation rate=methylated reads/ (methylated reads + unmethylated reads), *n* = 2 biologically independent samples, data are mean ± sd. **b** Violin plots of the distributions of DMRs in hypo- and hypermethylated groups. The median, maximum, and minimum values are shown. **c** Analysis of steroidogenesis-related genes. Fold change statistics for the FPKM value and methylation level differences between the ds*Dnmt1* and ds*CK* groups (ds*Dnmt1*/ds*CK*). Bar charts with padding means the methylation level was not detectable ("nd") at Up 2k or promoter regions, there are three biologically independent samples for gene transcription assays and two biologically independent samples for gene methylation assays. **d** Correlation analyses of 5mC changes and gene expression differences. The horizontal axis shows the difference in methylation (ds*Dnmt1*/ds*CK*), and the vertical axis shows the difference in gene expression

(log2 ds*Dnmt1*/ds*CK*). Red dots indicate upregulated genes, green dots indicate downregulated genes, and the number in each quadrant indicates the number of genes with the corresponding methylation and expression change patterns. **e** Eight TF genes were found by doing the intersecting between 413 hypo-methy and up DEGs and 259 trusted TF genes in *D. melanogaster* (*Dmel*). Fold change statistics for the FPKM value and methylation level differences of these eight TF genes were shown with bar chart, there are three biologically independent samples for gene transcription assays and two biologically independent samples for gene methylation assays. **f** *phm* relative expression levels were detected when top six TF genes with >2-fold upregulation were knocked down separately, *n* = 3 biologically independent samples. **g** 20E levels detected under dsRNAs treatment on day 9, *n* = 4 biologically independent samples. Data are mean ± sd, the differences were analyzed by two-tailed Student's *t*-test. Source data are provided as a Source Data file.

of follicle cells, excessive follicle cells dissociation was found after ds*Dnmt1* treatment (Fig. 6a). Moreover, the activity of cell death marker Caspase-3 was dramatically increased in follicle cells of ds*Dnmt1*-treated insects, and this ds*Dnmt1*-induced Caspase-3 activity could be blocked by the Pan Caspase inhibitor Z-VAD-FMK (Fig. 6b). These results suggest that the ds*Dnmt1*-treated follicle cells were likely undergoing anoikis, a cell death process related to detachment[38]. It is well-documented that FTZ-F1 induces the expression of matrix metalloproteinase (*Mmp*) genes, thus regulate the maturation and rupture of follicle cells in the ovaries of *Drosophila melanogaster*[39,40]. Consistent with this regulatory mechanism, *Mmp1* was upregulated in the ds*Dnmt1*-treated ovaries and rescued by ds*Dnmt1* and ds*ftz-f1* double RNAi (Fig. 6c). When *ftz-f1* was knocked down, the expression of *Mmp1* in the ovary was also significantly downregulated (Supplementary Fig. 4a). Additionally, no hypomethylation was observed in the Up 2k or promoter region of *Mmp1* (Supplementary Fig. 4b), indicating that *Mmp1* expression is not directly regulated by DNMT1-mediated 5mC modification. Moreover, the expression of five well-known chorion genes, *cathepsin-L*, *yellow-g*, *citrus*, *Fcp3c*, and *brownie*, in the transcriptome were explored[27,31,41]. There was no significant difference in their expression on day 7 PAE, and three of them (*yellow-g*, *citrus*, and *Fcp3c*) were significantly downregulated in the ds*Dnmt1* group on day 9 PAE (Supplementary Fig. 4c). The downregulation of these three genes was validated by qRT-PCR, while *cathepsin-L* was also found downregulated significantly when detected by qRT-PCR (Fig. 6d). Collectively, the hypomethylation-increased and prolonged *ftz-f1* expression results in the dissociation and anoikis of follicle cells, which might lead to the failure to synthesize sufficient chorionic proteins and further disrupt fertilization (Fig. 1h, i).

## Discussion

The composite data demonstrate that ds*Dnmt1* induces reproductive disability primarily through two mechanisms associated with *ftz-f1* overexpression: by causing aberrantly high levels of 20E, anoikis of follicle cells, and a shortage of chorion proteins. This impairment of choriogenesis eventually causes sponge-like body malformation and thus disrupts fertilization. In conclusion, 5mC modification prevents prolonged *ftz-f1* expression, ensuring timely choriogenesis and proper fertilization in the German cockroach (Fig. 7).

Identifying the key gene that tightly links DNA 5mC modification with insect reproduction in females is the primary challenge in the field. In a previous study on *O. fasciatus*, 5mC levels in the ovaries decreased after *Dnmt1* RNAi, but no quantitative changes in gene transcript expression were observed[7]. This negative result was likely due to inappropriate selection of the time point for RNA-Seq. In our study, we obtained a similar result with only a few DEGs on day 7 PAE, but numerous DEGs were found on day 9 PAE. In other words, the reduction in 5mC levels caused delayed developmental progression on day 9 PAE (Fig. 2a, b). Moreover, in both vertebrates and insects, there's an ongoing debate about the facilitative or preclusive effects of

DNA methylation on gene expression[42,43]. In this study, a large number of both upregulated genes and downregulated genes were observed after *Dnmt1* depletion, while the number of upregulated genes was much higher than downregulated genes (Fig. 2b), indicating that DNMT1-mediated DNA methylation mainly functions as a gene repressor in the German cockroach. Notably, an interesting question that arises now is how the expression of *Dnmt1* is precisely regulated during choriogenesis.

It has been reported that FTZ-F1 acts as a key factor involved in choriogenesis regulation[17], while this is the first instance where the gene has been shown to be controlled by 5mC modification regarding this reproductive process. *ftz-f1* expression was likely regulated by both positive and negative factors during choriogenesis: the 20E pulse should be responsible for inducing *ftz-f1* expression, consequently, DNMT1-mediated 5mC maintenance prevents prolonged *ftz-f1* expression. Interestingly, hypomethylation-induced aberrant elevations in FTZ-F1 levels overactivated the expression of steroidogenic genes and thus enhanced 20E levels in ovarian follicle cells, which disturbed the formation of sponge-like bodies and impaired fertilization. These results show that FTZ-F1 is involved in steroidogenesis regulation in ovarian follicle cells of *B. germanica*, and a similar relationship has been observed in the prothoracic glands of *D. melanogaster* larvae[35,44–46]. Additionally, exogenous 20E treatment only partially mimicked ds*Dnmt1*-induced phenotypic defects, which suggests more genes or factors are directly affected by ds*Dnmt1* treatment (Fig. 3d). Moreover, the regulation of female reproduction among insects might be regulated by multiple factors (e.g., JH, neuropeptides, and nutrition), and the epigenetic regulation might be involved in different physiological adaptations. We found that food deprivation downregulated the expression of *Dnmt1* (Supplementary Fig. 5), making it interesting to address these mechanisms further.

Previous studies have shown that FTZ-F1 can regulate follicle cell dissociation, choriogenesis, and fertilization by affecting the expression of downstream genes, including *Mmps*[39,40,47]. Cell dissociation is a classic marker for detachment-induced apoptosis, anoikis[38]. Here, we found that FTZ-F1 significantly promoted the expression of *Mmp1*, which might further lead to the cleavage of E-cadherin and result in excessive follicle cell dissociation eventually. The resulting follicle cell anoikis might be the reason for the abnormalities in the nuclei and cytoskeleton (Fig. 1I), leading to the failure to synthesize sufficient chorionic proteins (Fig. 6d). Moreover, FTZ-F1 has been found controlling the degeneration and death of the prothoracic gland by stimulating the expression of *E93* during the nymphal–adult transition in *B. germanica*[48,49]. *ftz-f1* was also reported playing a dual function in oviposition in the *B. germanica* recently; it promotes the expression of cytoskeleton and muscle-related genes at the early stage, while at the late stage, it maintains the expression levels of these genes[50]. In short, DNA 5mC modification plays key roles in follicle cell dissociation and choriogenesis. Of note, the effects of DNMT1 are likely genome-wide, and other DEGs or pathways except *ftz-f1* mentioned in this study

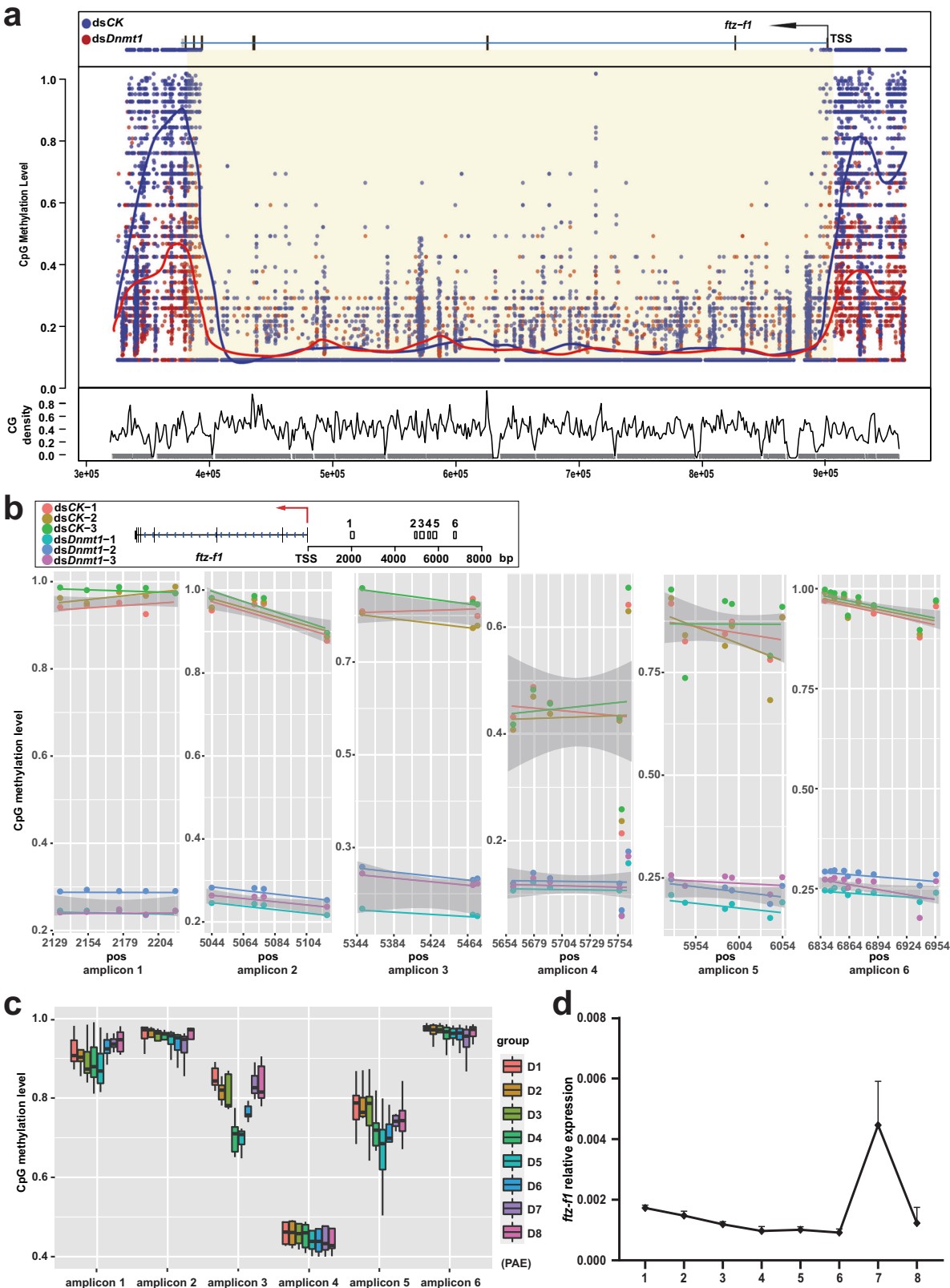

**Fig. 4 | *ftz-f1* promoter undergoes 5mC modification thus preventing prolonged expression. a** 5mC level changes of the *ftz-f1* in the genome detected by WGBS. The blue and red lines indicate the 5mC tendencies for the ds*CK* and ds*Dnmt1* groups, respectively. **b** 5mC level decreasing at the promoter region of *ftz-f1* was detected by NGS-BSP. Six amplicons were used and three replicates for each group were performed. **c** Boxplot results of temporal 5mC levels in the promoter region of *ftz-f1* during the first reproductive period (D1-D8) detected by NGS-BSP. Six amplicons were selected and three replicates for each group were performed. The median, maximum, and minimum values are shown. **d** Gene expression patterns of *ftz-f1* in the ovaries during the first reproductive period were detected by qRT-PCR, data are mean + sd, *n* = 3 biologically independent samples. Source data are provided as a Source Data file.

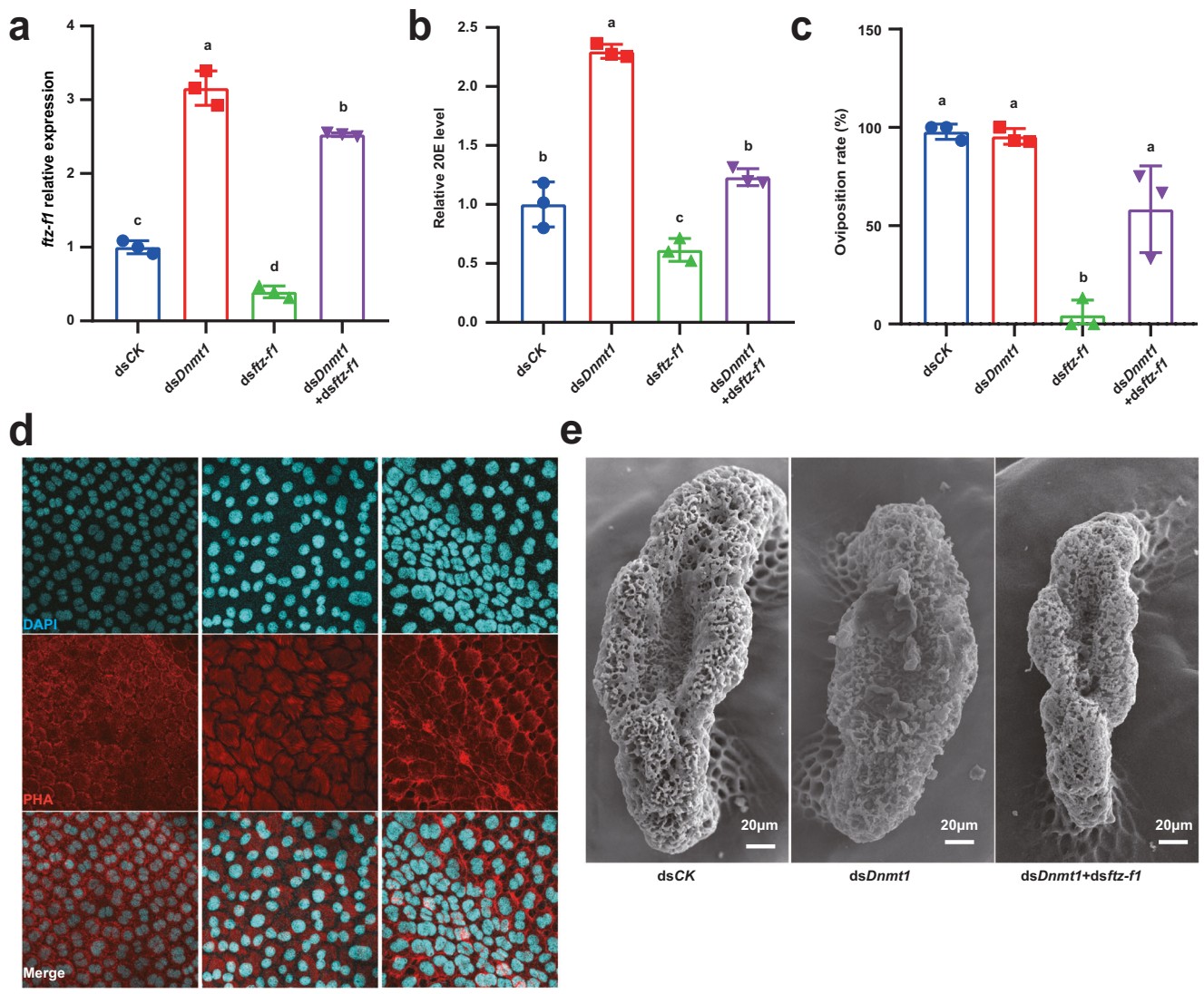

**Fig. 5 | dsDnmt1-induced phenomena can be partially rescued by ftz-f1 knockdown during choriogenesis. a** *ftz-f1* relative expression level under separate or combined injection of ds*Dnmt1* and ds*ftz-f1*. **b** 20E level detection under separate or combined injection of ds*Dnmt1* and ds*ftz-f1*. **c** Statistics of oviposit of female adults under separate or combined injection of ds*Dnmt1* and ds*ftz-f1*. **d** DAPI and PHA staining for nuclei and actin protein in ovarian follicle cells under separate or combined injection of ds*Dnmt1* and ds*ftz-f1*. **e** SEM observation of sponge-like bodies on eggs under separate or combined injection of ds*Dnmt1* and ds*ftz-f1*, 9 of 17 sponge-like bodies were observed normal. Data are mean ± sd, different letters indicate statistically significant differences between groups using one-way ANOVA and LSD multiple comparisons test, $P < 0.05$, $n = 3$ biologically independent samples. Source data are provided as a Source Data file.

might also be involved in the choriogenesis, because ds*ftz-f1* co-treatment only partially mimicked ds*Dnmt1*-induced phenotypic defects. For example, other unknown DEGs are likely involved in the reticular distribution of hollows on the tunica propria (Supplementary Fig. 1g).

Altogether, we propose a detailed regulatory mechanism by which DNMT1-mediated 5mC maintenance regulates female reproduction partially by repressing *ftz-f1* expression in *B. germanica*. Acting as a brake during normal choriogenesis, 5mC in the *ftz-f1* promoter regions prevents prolonged *ftz-f1* expression (Fig. 7a). Reductions in 5mC levels relieve the 5mC-mediated inhibition of *ftz-f1* expression during choriogenesis, and the aberrant elevations in *ftz-f1* expression disturb choriogenesis and disrupt fertilization by inducing an aberrantly high 20E level and causing excessive *Mmp1* expression, which in turn trigger anoikis of follicle cells and disrupt the formation of sponge-like bodies (Fig. 7b). This study provides a detailed regulatory mechanism by which 5mC modification precisely regulates insect reproduction.

## Methods

### Animal culture

The strain of *B. germanica* used in this study was originally collected from downtown Shanghai in the 1970s and is a well-established laboratory strain bred for nearly 50 years without exposure to insecticides. To maintain the colony, cockroaches were reared in plastic jars at 28 °C and 70% relative humidity in the dark. They were provided with rat chow and water[51]. Freshly emerged adults were separated from the colony after molting. The animals were randomly assigned to the test groups in all experiments. All animal protocols were reviewed and approved by the University Animal Care and Use Committee of South China Normal University.

### 5mC detection using dot blotting

For spatial comparative analysis of 5mC modification, six tissues, including the integument, head, thorax, legs, midgut, and ovaries, were dissected on day 5 PAE. For temporal analysis, the ovaries were collected daily throughout the first reproductive cycle (days 1-8 PAE).

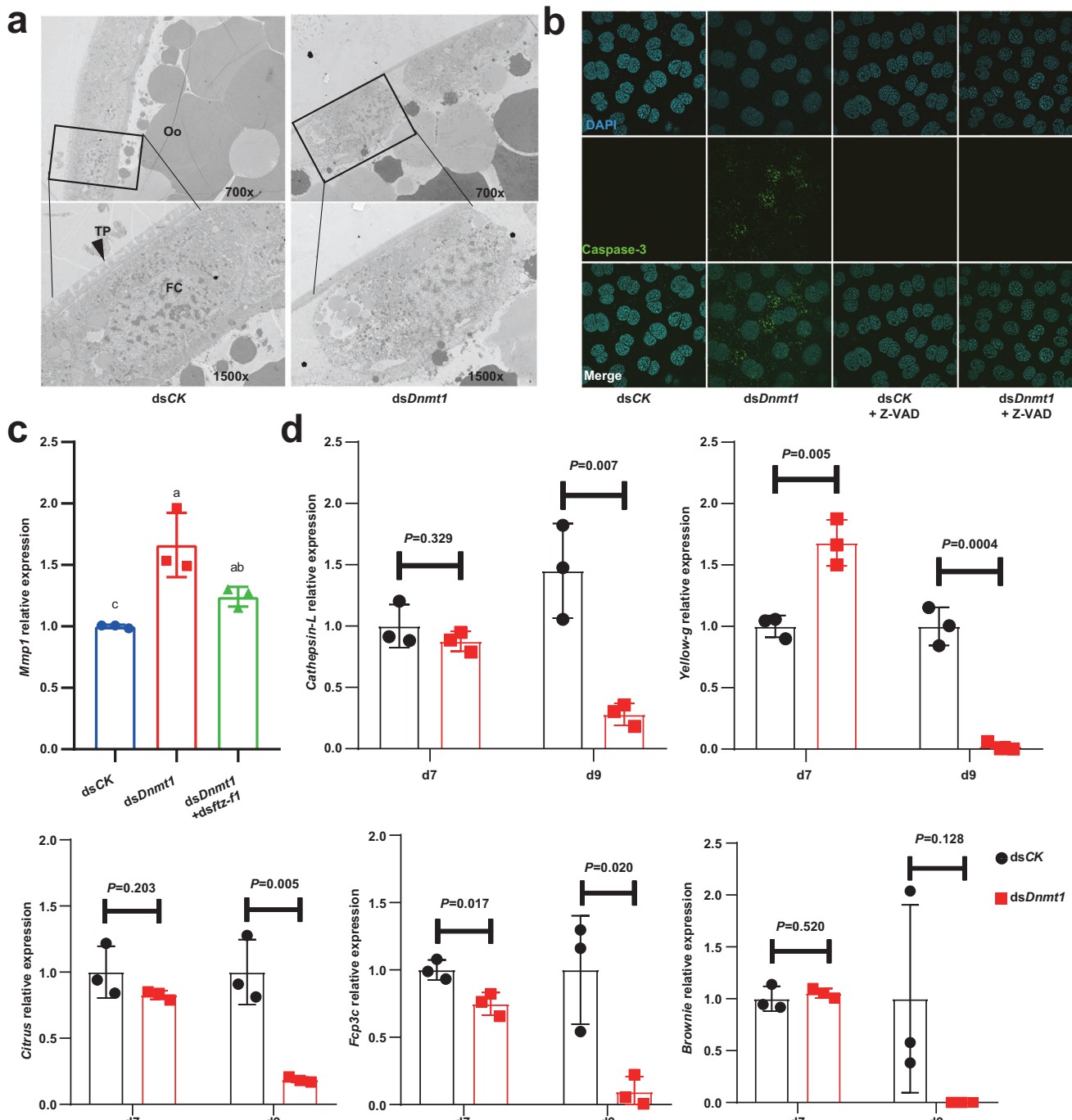

**Fig. 6 | 5mC modification prevents *ftz-f1* induced anoikis during cockroach choriogenesis. a** TEM observation of ovarian sections. TP: tunica propria; Oo: oocyte; FC: follicle cell. Two different magnifications, ×700 and ×1500, were selected. The asterisks indicate the cytomembrane detachment between follicle cells. **b** Apoptosis-associated Caspase-3 staining. Z-VAD means Pan Caspase inhibitor Z-VAD-FMK. **c** *Mmp1* relative expression level under separate or combined injection of ds*Dnmt1* and ds*ftz-f1*. The difference was analyzed by one-way ANOVA and LSD multiple comparisons test. Data are mean ± sd, different letters indicate statistically significant differences between groups, *P* < 0.05, *n* = 3 biologically independent samples. **d** The gene expression levels of five chorion genes detected by qRT-PCR at both d7 and d9 PAE. Data are mean ± sd, the significant difference was analyzed by two-tailed Student's *t* test, *n* = 3 biologically independent samples. Source data are provided as a Source Data file.

For 5mC detection after RNAi treatments, the ovaries were collected on day 5 PAE. Genomic DNA (gDNA) was extracted from the tissues using a gDNA extraction kit (Aidlab, DN14) and then treated with RNase to digest the RNA. gDNA on nitrocellulose membranes was incubated with a 1:5000 dilution of a 5mC-specific antibody (Abcam, 33D3) and then incubated with a 1:5000 dilution of an HRP-conjugated anti-rabbit IgG secondary antibody (Beyotime, A0239). The membranes were then treated with enhanced chemiluminescence reagent. The same amount of gDNA was loaded as input and stained with ethidium bromide dye at a 1:500 dilution (Thermo Fisher, 15585011).

### qRT-PCR for gene expression detection
For temporal analyses of *Dnmt1* gene expression in ovaries, tissues from three individuals were collected and mixed to form one sample at

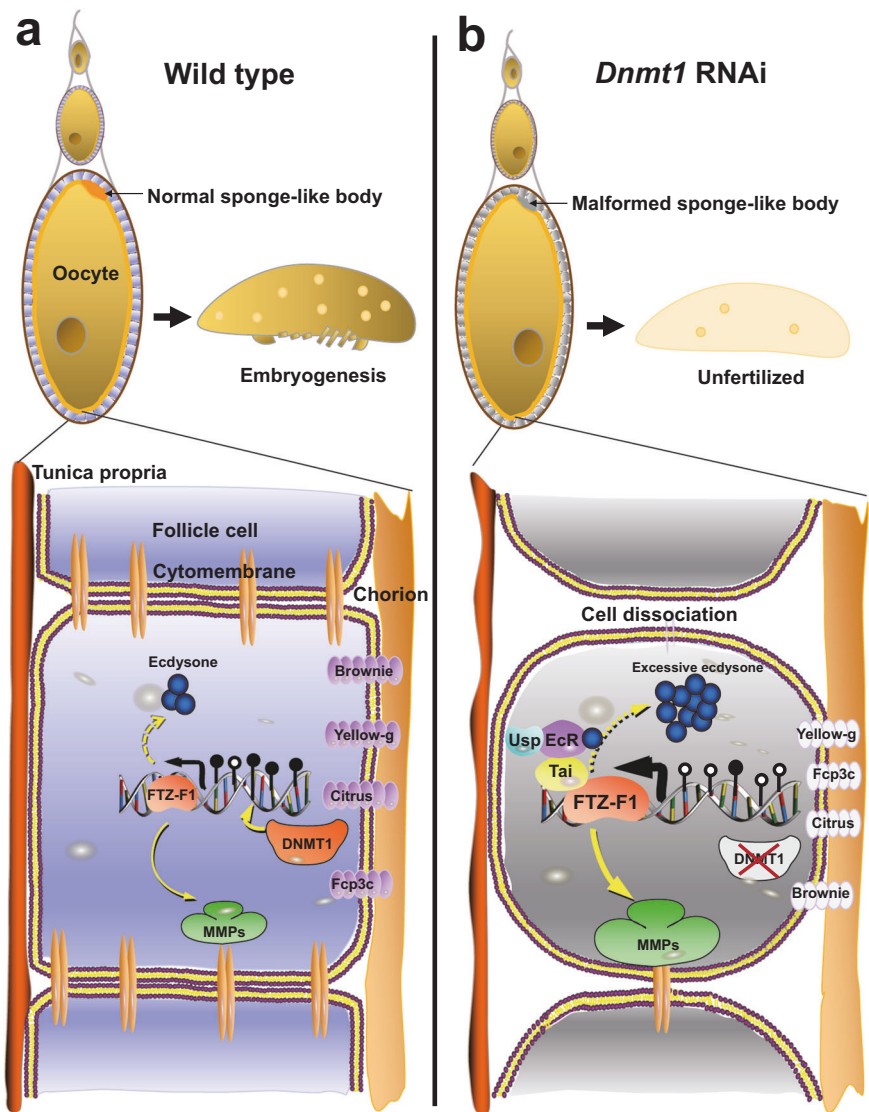

**Fig. 7 | Schematic view of the regulatory effect of DNMT1 on reproduction in *B. germanica*. a** In wild-type females (left panel), DNMT1-mediated DNA methylation in the promoter region of the *ftz-f1* was able to repress *ftz-f1* expression at suitable levels during choriogenesis. At its normal expression level, *ftz-f1* plays important roles in ovarian development, including the maturity of sponge-like bodies. **b** In ds*Dnmt1*-treated females (right panel), hypomethylation-induced *ftz-f1* upregulation impairs choriogenesis and disrupts fertilization by causing aberrantly high levels of 20E titer and 20E signaling, excessive follicle cell dissociation, and a shortage of chorion proteins.

each time point in the first reproductive cycle (days 1–8 PAE). For other gene expression analyses, ovaries from three individuals were collected after the corresponding treatments and were mixed to form one sample. Three replicates were tested for each gene. qRT-PCR was performed in triplicate using Hieff qPCR SYBR Green Master Mix (Yeasen, 11202ES03). The relative gene expression was calculated relative to *Actin-5c* expression using the ΔΔCt method following the manufacturer's instructions[14]. The primers used for qRT-PCR are shown in Supplementary Table 1, and all the specificities of the primers were confirmed by Sanger sequencing of the PCR products.

**Double-stranded RNA treatment**

For the negative control, ds*CK* targeting a clone vector sequence was used[15]. Sense and antisense RNA was synthesized in a single reaction using a T7 RiboMAX Express RNAi System (Promega, P1700). After purification, the double-stranded RNA (dsRNA) concentration was adjusted to 3 µg/µl[15]. Under normal conditions, female adults were injected with 6 µg of ds*Dnmt1* between the abdominal sternites using an insulin syringe for each injection. To thoroughly disrupt gene

function, three injections were performed (on days 2, 4, and 6 PAE). The *Dnmt1* RNAi efficiency was detected on day 5 PAE after twice injection on day 2 and day 4 PAE. For transcript factor genes *hkb*, *ftz-f1*, *Hr39*, *Mi-2*, *bab2*, and *CrebA*, RNAi efficiency was detected on day 5 PAE after twice injection on day 2 and day 4 PAE. For *phm*, *tai*, and *EcR*, 6 µg of double-strand RNAs were injected on day 7 PAE, and relative expression levels were detected on day 9 PAE. In the double RNAi experiments, 6 µg of ds*CK* or ds*Dnmt1* were injected on day 2, 4, 6 PAE, and ds*ftz-f1* were injected on day 7 PAE with 10 ng. All dsRNA-injected female adults were paired with wild-type male adults for mating. The primer sequences for dsRNA synthesis can be found in Supplementary Table 2.

**LC–MS/MS detection of DNA 5mC modification**

DNA 5mC modification was detected by MetWare Company (Wuhan, China) with an AB Sciex QTRAP 6500 LC–MS/MS platform. The ovaries were dissected from female adults treated with ds*CK* or ds*Dnmt1*, and at least three replicates were tested for each group. gDNA was digested by *Dpn*I (NEB, R0176S) to remove possible bacterial DNA.

gDNA completely digested with deoxyribonuclease was analyzed via LC–MS/MS. The ESI–MS/MS instrument used was an AB 6500 QTRAP LC–MS/MS system equipped with an ESI Turbo Ion-Spray interface operating in both positive and negative ion modes and controlled by Analyst 1.6.3 software (AB Sciex).

## Whole-genome bisulfite sequencing analysis of cytosine methylation

For detection of 5mC levels in *B. germanica* after ds*CK* or ds*Dnmt1* treatment, ovaries at day 9 PAE that had been treated with dsRNA were dissected; two biological replicates were tested for each treatment. dsRNA injections were performed on days 2, 4, and 6 PAE (three times), and ovaries from six individuals were dissected and mixed to form one sample. The gDNA was extracted for WGBS library construction using an AceGen Bisulfite-Seq Library Prep Kit (AceGen, AG0311) according to the manufacturer's protocol. The gDNA from ovaries was extracted for WGBS library construction. After bisulfite treatment, the DNA was amplified in 10 cycles of PCR with Illumina 8 bp dual index primers. The constructed WGBS libraries were then analyzed with an Agilent 2100 Bioanalyzer and finally sequenced on Illumina platforms using a 150×2 paired-end sequencing protocol. The FastQC tool was used to perform basic statistical analysis on the quality of the raw reads. BSMAP software was used to map the bisulfite sequences to the reference genome. The methylation levels of individual cytosines were calculated with the formula ML = mC/(mC+umC), where ML is the methylation level and mC and umC represent the numbers of reads supporting methylated C and unmethylated C, respectively. Differentially methylated regions (DMRs) were identified using metilene (ver. 0.2–7)[52], and the regions with Kolmogorov–Smirnov (KS)-test *P* value < 0.05 and Benjamini & Hochberg (BH)-corrected *P* values < 0.05 were considered DMRs. Based on the results of DMR annotation and Kyoto Encyclopedia of Genes and Genomes (KEGG) annotation[53], functional enrichment analysis was performed on genes whose gene bodies or 2 kb upstream or downstream regions overlapped with a DMR.

## Tissue staining

For ovary staining, fresh ovaries were fixed in 4% paraformaldehyde in PBS for 2 h, and then washed in PBS supplemented with 0.3% Triton (1×PBT). For DAPI (Beyotime, C1002) and phalloidin (PHA) (Yeasen, 40734ES75) staining, the ovaries were incubated with a 1:1000 dilution of phalloidin in 1× PBT for 2 h, and 1 μg/ml DAPI was added for the last 5 min. Three washes with 1× PBT were performed before observation. For egg pronucleus staining, the processes were performed based on modified CUBIC protocol[54]. The pronucleus was stained by primary α-tubulin antibody (Sigma, MAB1864-I), and Cy3-labeled Goat Anti-Rat IgG(H + L) (Beyotime, A0507) was used as the secondary antibody. For Caspase-3 activity detection, the ovaries were incubated with a 1:2000 dilution of a Caspase-3 (Beyotime, AC033) rabbit monoclonal antibody overnight at 4 °C. After three washes with 1× PBT, the ovaries were incubated with a 1:1000 dilution of an Alexa Fluor 488-labeled goat anti-rabbit IgG (H + L) (Beyotime, A0423) secondary antibody. All the signals were observed via confocal microscopy.

## Scanning electron microscopy

Selected eggs from ds*Dnmt1*- and ds*CK*-treated female adults were fixed in 2.5% glutaraldehyde and 2% paraformaldehyde in cacodylate buffer (0.2 M) for at least 2 h. After rinsing twice with the same buffer, the samples were treated with 1% osmium tetroxide (Ted Pella) at 4 °C for 1 h. The tissues were dehydrated with increasing concentrations of ethanol at 15 min intervals. Finally, the samples were subjected to critical-point drying to complete the dehydration process. The samples were attached to stubs with double-stick tape, coated with gold-palladium in a sputter coating apparatus, and then observed via scanning electron microscopy (SEM) at 5 kV (JEOL JSM-6360LV).

## Transmission electron microscopy

Ovarian follicles were isolated from both ds*Dnmt1*- and ds*CK*-treated female adults at the end of the choriogenic period and fixed in fixative for transmission electron microscopy (TEM; Servicebio, G1102) for at least 2 h. After rinsing three times with PBS, the samples were treated with 1% osmium tetroxide (Ted Pella) in 0.1 M PBS (pH 7.4) at room temperature for 2 h. The osmium tetroxide was removed by rinsing in 0.1 M PBS three times. The tissues were dehydrated with increasing concentrations of ethanol at 15 min intervals and infiltrated with increasing concentrations of EMbed 812. Ultrathin sections (60–80 nm) were cut with an ultramicrotome. The sections were stained with uranyl acetate and lead citrate in a stepwise manner. Finally, the sections were air-dried overnight at room temperature and then observed with a transmission electron microscope (Hitachi, HT7700)[55].

## RNA-Seq and data analysis

Ovaries at day 7 (d7) and day 9 (d9) PAE that had been treated with dsRNAs were dissected. Three biological replicates were tested for each treatment. The dsRNA injections were performed on days 2, 4, and 6 PAE (three times), and ovaries from six individuals were dissected and mixed to form one sample. A total of 1 μg of RNA per sample was used as input material for RNA preparation. Sequencing libraries were generated using an NEBNext Ultra™ RNA Library Prep Kit for Illumina (NEB, E7770S) following the manufacturer's recommendations. The clustered libraries were sequenced on an Illumina platform, and paired-end reads were generated. Raw FASTQ-format data were first processed through in-house Perl scripts. HISAT2 software was used to map the reads to the reference genome (*B. germanica* Bger_1.1). Gene expression levels were estimated as fragments per kilobase of transcript per million fragments mapped (FPKM) values, and differential expression analysis between the two groups was performed using DESeq. Genes with adjusted *P* values < 0.05 in DESeq were considered as differentially expressed. KEGG pathway enrichment analysis was performed for the differentially expressed genes using the R package clusterProfiler v4.8.3.

## 20E level detection

Fresh ovaries were dissected on day 9 PAE and extracted with 200 μl of 100% methanol. The tissues were homogenized and centrifuged at 13000 × *g* for 5 min, and the supernatant was collected in a new tube. The remaining precipitate was resuspended with another 200 μl of methanol and centrifuged again to collect the supernatant[56]. The supernatants were mixed together and air-dried with nitrogen to collect the 20E. The 20E concentration was detected with 20E ELISA kit according to the manufacturer's instructions (Mlbio, ml062795).

## Exogenous 20E and nutrition-related components treatment

For 20E treatment, newly emerged female adults were injected with 1 μl of 0.5 μg/μl 20E (10% ethanol) four times on days 2, 4, 6, and 8 PAE. 1 μl of 10% ethanol solution was injected as a negative control.

For the nutrition-related components (insulin, Aa mixture, and glucose) treatment, bovine insulin (Yuanye Bio-Technology, 11070-73-8) was solubilized in DMSO at a concentration of 25 μg/μl; the Aa mixture (type H) (Wako, 013-08391) was diluted with PBS solution into 12.5 μg/μl; the glucose (Beyotime, ST1024) was diluted with PBS into 25 μg/μl. 2 μl of different kinds of components and their corresponding solvents were injected into the abdomen of female adults on day 5 PAE. All injected females were paired with wild-type male adults for mating. The eggs were collected at 84 h post oviposition.

## Next-generation sequencing-based bisulfite sequencing PCR

*ftz-f1* promoter region-specific DNA methylation was assessed by next-generation sequencing-based bisulfite sequencing PCR (NGS-BSP).

In brief, six pairs of BSP primers were designed using the online MethPrimer software and were listed in Table S3. One microgram of genomic DNA was converted using a ZYMO EZ DNA Methylation-Gold Kit (Zymo, D5005), and one-twentieth of the elution product was used as the template for PCR amplification with 35 cycles using a KAPA HiFi HotStart Uracil⁺ ReadyMix PCR Kit (Kapa, KK2801). For each sample, the BSP products of multiple DMRs were pooled equally, 5′-phosphorylated, 3′-dA-tailed, and ligated to a barcoded adapter using T4 DNA ligase (NEB, M0202S). The barcoded libraries from all samples were sequenced on the Illumina platform.

## Statistics and reproducibility
The hatchability experiment of *B. germanica* was repeated three times, with 15 individuals per group treated. The mass spectrometry and quantitative experiments had three biological replicates involving 12 individuals. The data statistics in the paper were analyzed by the software SPSS 25, and the specific analysis method is shown in the figure legends.

## Reporting summary
Further information on research design is available in the Nature Portfolio Reporting Summary linked to this article.

## Data availability
The RNA-Seq data and WGBS data generated in this study have been deposited in the Sequence Read Archive (SRA) database under public accession code PRJNA700816. The original data generated for plots and charts in this study are provided in the Source Data file. Source data are provided with this paper.

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

## Acknowledgements

We would like to thank Dr. David Jay Segal and Xavier Belles for the critical reading of the manuscript. This work was supported by the National Natural Science Foundation of China (Grant Nos. 32220103003, 31930014 to S.L, 32370439, 32070500 to C.R., and 32000334 to D.Y.), by the Laboratory of Lingnan Modern Agriculture Project (Grant No. NT2021003 to S.L.), by the Natural Science Foundation of Guangdong Province (Grant No. 2021B1515020044 to C.R.), by the Department of Science and Technology in Guangdong Province (Grant Nos. 2019B090905003 to S.L.), by the Shenzhen Science and Technology Program (Grant No. KQTD20180411143628272 to S.L.).

## Author contributions

S.L. and C.R. conceived the project. C.R. and S.L. designed and led the project. Z.Z. and L.Li performed most of the experimental work and analyzed the data. R.Z., L.Lin, M.L., Y.W., N.L., and Y.C. also performed functional experiments. M.L., D.Y., and S.Z. conducted bioinformatic data analysis. C.R., S.L., and Z.Z. wrote the manuscript. Z-M.Z improved the manuscript.

## Competing interests

The authors declare no competing interests.
