## [Peer Review File · Nature Communications]

5mC modification orchestrates choriogenesis and fertilization by preventing prolonged Ftz-f1 expressionREVIEWER COMMENTS

Reviewer #1 (Remarks to the Author):

Zhao et al. report that Dnmt1 depletion in the German cockroach resulted in a reduction of DNA methylation on cytosine (5mC modification), as checked in the ovaries. Moreover, Dnmt1 depletion triggered an increase of expression of a high number of genes during choriogenesis. Among them the authors chose *ftz-f1* to be further studied, a gene involved in ecdysteroidogenesis, as well as in ecdysone signalling. Additionally, the authors observed that DNA methylation on cytosine was reduced at the *ftzf1* promoter region. At the same time, an increase and prolongation of *ftz-f1* expression in the ovarian follicle cells during choriogenesis was observed, and abnormally high levels of 20E were measured. This resulted in different effects, namely impairment of choriogenesis, especially in the correct formation of the sponge like body (which is essential for the entry of spermatozooids), up-regulation of the extracellular matrix gene *Mmp1*, and anoikis and death of the follicle cells. An obvious final consequence is that fertilization was prevented as the sponge-like body became non-functional. Considering all these data, the conclusion that emerges is that timely DNA methylation at the *ftzf1* promoter reduces the expression of this gene, as seen in the ovary, which is crucial to correctly culminate choriogenesis.

The biological effects of DNA methylation are still under debate, even at the basic level of discerning whether it facilitates or precludes gene expression. This study shows how DNA methylation is capable of regulating hormone production (by decreasing the expression of the key gene *ftz-f1*) and the chorionation of oocytes in insects. This result is important not only in the context of the specific issue of oocyte chorionation, but also in the context of the general debate about the effects of DNA methylation and gene expression. This makes the manuscript interesting to an audience larger than that of insect molecular physiologists.

Despite the fact that the information provided comes from separate evidence, all the concatenated results clearly point to the above conclusion, the one claimed by the authors. The methodology is diverse and powerful, it is sufficiently described, and it is adequate to answer the successive questions that arise throughout the work. I see no technical or methodological problems in their use.

The manuscript, however, can be improved, and the following comments could help to do so.

General, fundamental comments

1. The analysis of what happened to the chorion after the depletion of DNMT1 and the increase and prolongation of *ftz-f1* expression can be further addressed by measuring the expression of chorion genes from different stages of choriogenesis (described in doi: 10.1186/1471-2164-10-206). For example, the authors could measure the expression of cathepsin-I (very early choriogenesis), yellow-g (early choriogenesis), citrus (early-mid choriogenesis), brownie (late choriogenesis), to see approximately when the process fails.

2. Analysis of transcriptomes. The effort made in the preparation and analysis of the transcriptomes deserves to be exploited more efficiently and specifically. The authors could go deeper into the DEGs, identifying which chorion genes are differentially expressed, and validating the transcriptomic values by qRT-PCR. This would help to disambiguate expressions like "leading to the failure to synthesize sufficient chorionic protein" (line 283), and have a general view of chorion-related genes affected by the experiment.

3. Structure of the manuscript and Discussion. The Results section is complex, but the structure seems adequate, each section leading naturally to the next. In contrast, the Discussion looks poor and redundant. This section should be revised, in terms of discussing the results in relation to other contributions, not only in *Blattella* (there are a good number of contributions that can be compared with the results of Zhao et al.) but also in other insects. The Discussion could also address the debate on how DNA methylation affects gene expression, considering not only insects (doi: 10.1093/molbev/msw264), but animals in general (see, for example, doi: 10.1186/s13072-018-0205-1). The authors can also comment on possible future directions. For example, an interesting question that arises now is to figure out what determines the increase in DNMT-1 expression precisely during choriogenesis.

General formal comments

4. Nomenclature of fushi tarazu factor 1. Please use the full name in the first mention. Then abbreviate it as *ftz-f1* (in italics, with lowercase initial, not capitalized as seen in most sections of the manuscript) when referring to the gene, and as FTZ-F1 when referring to the protein.

5. Teleological expressions. I refer to expressions like “is a crucial epigenetic modification FOR regulating gene expression” (line 23-24), or “Ftz-f1 promoter undergoes 5mC modification TO prevent prolonged expression” (line 202), or “induces the expression of matrix metalloproteinase (Mmp) genes TO regulate the maturation and rupture of follicle cells” (lines 241-242) (capitals are mine). These expressions sound teleological, that is, they denote purpose, and there is no purpose in developmental and evolutionary biology processes, there is simply natural selection. It is easy to avoid the "FOR" or "TO" using alternative wording. For example: “is a crucial epigenetic modification INVOLVED IN regulating gene expression” (line 23-24), or “Ftz-f1 promoter undergoes 5mC modification THUS preventing prolonged expression”, or “induces the expression of matrix metalloproteinase (Mmp) genes THUS regulating the maturation and rupture of follicle cells”.

6. Background on *Blattella* studies. The authors can be interested in reading papers that are relevant to the work presented. For example, about vitellogenesis: DOI: 10.1111/j.1365-3032.1995.tb00801.x; DOI: 10.1016/s0965-1748(99)00058-2; DOI: 10.1111/j.1742-4658.2005.05066.x.; DOI: 10.1016/0012-1606(86)90143-0. About choriogenesis: DOI: 10.1016/0040-8166(93)90019-h; DOI: 10.1186/1471-2164-10-206; DOI: 10.1016/j.ibmb.2010.11.001. About epigenetics and choriogenesis: DOI: 10.1016/j.yexcr.2013.07.006. About *ftz-f1*, choriogenesis, cell death: doi.org/10.1111/imb.12866; DOI: 10.1016/j.ydbio.2010.07.012; DOI: 10.1242/dev.190066. Etc.

Specific comments

7. The sentence “Reproduction is a fundamental process for all known organism, as each individual organism is the result of reproduction” (line 52-53) is trivial and unnecessary. The next one can perfectly serve as the initial sentence of the section.

8. Line 55. “production of yolky eggs”, read “production of yolky oocytes”. “laying of fertilized eggs”, read “laying of eggs”. “As the organs responsible for egg production (oogenesis)”, read “As the organs responsible for oogenesis”.

9. Line 63. "In most, if not all, insects, 20E regulates the latest stage of oogenesis, choriogenesis 16, 17". A relevant work in *Blattella*: DOI: 10.1016/0040-8166(93)90019-h

10. Line 78. "Female reproduction and its regulation have been well studied in the German cockroach 23.". Reference 23 by itself is not representative of the extensive work published on *Blattella* reproduction. See my comment 6 above for additional information.

11. Line 110. "Abundant energids in the yolk mass were observed in the dsCK-treated eggs, which tended to concentrate at the ventral surface at 60 h and developed into early embryos at 84 h 24.". Reference 24 does not describe this. This is described in DOI: 10.1016/j.ibmb.2010.04.006 and in DOI: 10.1016/j.isci.2018.05.017

12. Line 164. "PAE. In accordance with the situations in other species, methylation modifications and changes were found at the cytosines of the CpG motif". This has been reported specifically in *Blattella*: DOI: 10.1016/j.isci.2020.101778

13. Line 188. "the transcription factor gene ftz transcription factor 1 (Ftz-f1)". See my comment 4, above.

14. Line 199. "These findings suggest that FTZ-F1 acts not only as a transcription factor promoting steroidogenesis but also as a competence factor that works in conjunction with Tai to enhance EcR activity for 20E signalling 29, 30.". This cannot be inferred from the empirical evidences reported.

15. Line 220 "The co-treatment of dsDnmt1 increased the depleted Ftz-f1 expression caused by dsFtz-f1 treatment. In other words, the co-treatment of dsFtz-f1 attenuated the increased and prolonged Ftz-f1 expression caused by dsDnmt1 treatment (Fig. 5A)." This is quite confusing. One (good and straightforward) sentence would be enough.

16. Line 224. "Since dsFtz-f1 treatment can completely disrupt SPAWNING process, the co-treatment of dsDnmt1 increased the SPAWNING rate back..." (capitals are mine). Spawning does not exist in cockroaches.

17. Line 228. "These genetic interaction". Ambiguous. What genetic interactions are the authors referring to?

18. Line 249 to 254. This paragraph fits better in the Discussion section.

19. Line 266. "the 20E PURGE should be responsible for inducing Ftz-f1 expression" (capitals are mine). I presume the authors mean "20E pulse".

20. Line 283. "leading to the failure to synthesize sufficient chorionic protein". There are well characterized chorion genes well characterized in *Blattella*, like citrus (early chorion), brownie (sponge like body) etc. (see my point 1 above) that could be measured. The detailed analysis of the transcriptomes could also reveal chorion proteins that are "insufficient" (see my point 2 above).

21. Line 317 and the following. Please indicate what is the unit of measurement in quantitative gene expression (I presume copies per copy of actins)

22. Line 433. *Blattella germanica* must be in italics.

23. Reference 11. Instead of citing a book review, the authors might cite the book itself.

24. Figure 1A. “Throax”. Read “Thorax”.

Reviewer #2 (Remarks to the Author):

Here, Zhao et al aim to unravel how CpG methylation regulates female reproduction in the cockroach *Blattella germanica*. In their manuscript, Zhao et al present two stories. First, they provide RNAseq of DNAmethylase (Dnmt1) knock-down and control ovaries at days 7 and 9 post adult emergence, as well as whole-genome bisulphite sequencing of Dnmt1 knockdown and control ovaries at day 9. Second, they provide a functional study of the candidate genes *Ftz-f1* and *Mmp1* in ovary development, including analysis of the methylation pattern of the two candidate genes. The first story would have been suitable for publication in Nature Communications if an unbiased approach in identifying pathways and candidate genes would have been taken. The second study is well performed, but may be more suitable for a specialized journal. The main problem with the manuscript is that these two stories are quite unrelated, or at least: the second story does not at all logically follow from the first, as I explain in the three paragraphs below, and conclude in the fourth paragraph.

To start, it is a bit unclear what the main question of the paper is. In the abstract, the authors write: “DNA methylation...is a crucial epigenetic modification for regulating gene expression, but little is known about its regulatory function and the underlying mechanism in insects” (lines 23-25), and: “This study significantly advances the understanding of how DNA 5mC modification regulates...female reproduction in insects” (lines 35-36). In the introduction, the authors write: “However, there is limited knowledge regarding the mechanistic links between 5mC modification and insect development, particularly regarding transcriptional regulation” (lines 48-50), and: “...we...identified how 5mC modification regulates key aspects of female reproduction” (lines 81-82). Although no single clear question is presented, the reader now gets a feeling of what the authors want.

The reader now expects things such as knockdown of methylating enzymes, bisulphite sequencing and RNAseq, exactly like the authors present, followed by an unbiased functional analysis of the most outstanding and promising pathways and candidate genes. The key problem is that the authors do not do the latter. When they analyse the RNAseq data in line 146, the authors state out of the blue: “Our analysis revealed that the insect hormone biosynthesis pathway was enriched (Fig. S2).” But when I look at Figure S2, “Insect hormone signaling” is only a minor category. Why do the authors choose this category for further study? There are many more and much higher over-represented pathways, such as ubiquitination, endocytosis, or wnt signaling. Why not going into these? The authors give no explanation, and this is the main problem of the manuscript. They go on in lines 146-148: “Further examination of the upregulated genes in this pathway identified phantom (*Phm*) and shadow (*Sad*), both of which are involved in steroidogenesis (Fig. 2C)”. Yes, they now specifically look at *Phm* and *Sad* and show mild upregulation. Much more interesting would be to show where *Phm*, *Shd* (or the later analyzed *Ftz-f1* and *Mmp1*) are in the volcano plot in Figure 2B. Do they belong to the genes that show the largest upregulation? I don't think so. What are actually the most outstanding dots in Figure 2B? This would be a way to

objectively identify functionally involved genes in an unbiased way.

The same happens when the authors add the bisulphite sequencing data. Out of the blue, the authors write in lines 188-189: “Subsequently, we analyzed the hypomethylated and upregulated genes in top-left quadrant and identified the transcription factor gene *ftz* transcription factor 1 (*Ftz-f1*)...”. Which of these dots in the upper left quadrant of Figure 3C actually is *Ftz*? Does it stand out? What genes do stand out? Again, the choice for further study seems biased. To make the problem complete, *Phm* and *Sad* do not show any differential methylation, and neither does the later studied *Mmp1* (Fig. S4). Hypomethylation of the main gene *Ftz* is later shown by targeted bisulfate sequencing PCR, but *Ftz* does not stand out in the original whole-genome approach.

In conclusion, the authors provide a very good study on how *Ftz* and ecdysone are involved in female ovary development and oogenesis. They do show that methylation of *Ftz* is involved. This is a lot of work and highly interesting for a more specialized journal. However, the authors fail to provide an unbiased analysis of targets of methylation, and are far from elucidating how methylation “precisely regulates insect reproduction”.

Experimental issues:

-In figure 1e, the authors show that fertilization does not take place upon *Dnmt1* RNAi. This is of course crucial. Otherwise, it could be that the embryos start to develop, but that failure of embryonic development is due to *Dnmt1* knockdown in the embryo, i.e. aberrant methylation during embryonic development. On how many eggs is this conclusion based? Same question for Figure 1F: how many eggs have been analysed? And for Figure 1I: how many follicles have been analysed? N should be crystal clear and indicated in the figure legend.

-Lines 154-155: “while the mating and ootheca-laying rates of the insects were unaffected”
Where are the data that show this?

-Lines 156-157: “most of the sponge-like bodies in the 20E-treated animals were malformed”
How many sponge-like bodies have been investigated, and how many were malformed?

-Figure 3B: Why do we see an increase in hypermethylation upon *Dnmt1* RNAi? Isn't this unexpected, as one would expect a decrease in methylation when a methylation enzyme is knocked down?

-Line 224: I am not a native speaker, but “spawning” suggest to me something organisms can only do in an aquatic environment. What do the authors mean? How is the “spawning rate” in Figure 5C determined?

Minor comments:

-I wonder why the authors consistently speak about “5mC modification” (e.g. title and lines 72 and 81). It is of course actually the Cytosine that is modified, and the product is 5mC. Sometimes, the authors seem to use “5mC” to refer to the process of methylation, e.g. lines 23 and 41: “DNA methylation on cytosine (5-methylcytosine, 5mC)...”, whereas the process is of course methylation; the result is 5mC. Why not simply calling it CpG methylation, as

many other papers do? The authors nicely prove that it is indeed CpG methylation, and not any other sites such as CHG or CHH (line 166).

-Maybe the authors could mention in the legend of Figure 1a what the staining for "input" is.

-Fig1a': does oviposition take place after day 8?

-Fig. 1e: what do the small pictures show? Please explain in the legend.

-Lines 110-115: could the authors provide references in the text to the exact sub-panel of Figure 1F that show what the authors describe. Maybe some arrows or a better description in the legend would help the reader.

-Figure S1F: "the main component Vg protein in the oocytes was marked". What do the authors mean? That the thickest band is Vg? How do the authors know?

-Lines 142-143: "These transcriptomic changes were consistent with the morphological changes observed at both time points" In what sense? That there is no effect on day 7, but a clear effect on day 9?

-Figure S3D: What does Dmel mean? Demethylated? As I also see Bger (which probably means *Blatella germanica*), it could also be *Drosophila melanogaster*? Please explain the legend. What is actually analysed and shown here? Are only transcription factors taken into account? Why?

-Line 198: "Phm, Ftz-f1, EcR, or Taiman" Why were these genes chosen? What kind of gene is Taiman? Taiman comes out of the blue and could use some introduction.

-Line 205: "the comprehensive 5mC level around the Ftz-f1 gene in the genome was checked". How? I guess looking at the whole-genome bisulfite sequencing data?

-What is the size of the Ftz gene/locus? Figure 4B suggests that the gene is around 8kb, i.e. an order of magnitude that can be expressed as 10^3 bp. The x-axis and gene model on panel 4A, however, suggest an order of magnitude of 10^5 bp.

-Figure 4C: These 6 amplicons are different from the 6 amplicons in Figure 4B. Where exactly are these amplicons of panel 4C located? Maybe the authors can draw something similar like the schematic drawing in 4B. And how do the authors define "the promotor" in line 209? Simply the 2kb upstream sequence?

-Lines 235-237: could the authors help the reader a bit with what should be seen in the electron micrographs? (maybe some arrow: where do we see dissociation?)

-Line 237: why are the authors suddenly going to look at caspase3? For the general audience of Nature Communications, the function of caspase3 should be introduced and a rationale to study this gene should be given.

-Line 446: Who of the authors is E.D. ?

Reviewer #3 (Remarks to the Author):

I think the present investigation clarified the role of 5mC modification on chorionation. It also confirmed the inhibitory mechanism by regulating Ftz-F1 expression. The methodology is sound and the results are convincing. I therefore support publication of this manuscript after minor English correction. I am curious how dnmt1 will be turned on or regulated upon starvation that induces oosorption. Comments of this question may be desirable.

** Please also note that Reviewer 3 has included a marked-up manuscript, which is attached to this message.

Response to reviewers
5mC modification orchestrates choriogenesis and fertilization by
preventing prolonged *ftz-f1* expression
(NCOMMS-23-31301A)

We thank the reviewers for providing constructive feedback, and we appreciate the positive and critical comments on this manuscript. We have fully revised our manuscript and have addressed all the points raised by the reviewers, and we think the resulting manuscript is much improved. Specific revisions and point-by-point responses are provided below.

REVIEWER COMMENTS

Reviewer #1 (Remarks to the Author):

Zhao et al. report that Dnmt1 depletion in the German cockroach resulted in a reduction of DNA methylation on cytosine (5mC modification), as checked in the ovaries. Moreover, Dnmt1 depletion triggered an increase of expression of a high number of genes during choriogenesis. Among them the authors chose *ftz-f1* to be further studied, a gene involved in ecdysteroidogenesis, as well as in ecdysone signalling. Additionally, the authors observed that DNA methylation on cytosine was reduced at the *ftzf1* promoter region. At the same time, an increase and prolongation of *ftz-f1* expression in the ovarian follicle cells during choriogenesis was observed, and abnormally high levels of 20E were measured. This resulted in different effects, namely impairment of choriogenesis, especially in the correct formation of the sponge like body (which is essential for the entry of spermatozooids), up-regulation of the extracellular matrix gene *Mmp1*, and anoikis and death of the follicle cells. An obvious final consequence is that fertilization was prevented as the sponge-like body became non-functional. Considering all these data, the conclusion that emerges is that timely DNA methylation at the *ftzf1* promoter reduces the expression of this gene, as seen in the ovary, which is crucial to correctly culminate choriogenesis.

The biological effects of DNA methylation are still under debate, even at the basic level of discerning whether it facilitates or precludes gene expression. This study shows how DNA methylation is capable of regulating hormone production (by decreasing the expression of the key gene *ftz-f1*) and the chorionation of oocytes in insects. This result is important not only in the context of the specific issue of oocyte chorionation, but also in the context of the general debate about the effects of DNA methylation and gene expression. This makes the manuscript interesting to an audience larger than that of insect molecular physiologists.

Response: Thanks for your time taken to review our manuscript. We are very grateful for your high praise that our work will obtain the interest from a broad audience.

Despite the fact that the information provided comes from separate evidence, all the concatenated results clearly point to the above conclusion, the one claimed by the authors. The methodology is diverse and powerful, it is sufficiently described, and it is adequate to answer the successive questions that arise throughout the work. I see no technical or methodological problems in their use. The manuscript, however, can be improved, and the following comments could help to do so.

Response: Thanks again for your constructive comments. We have revised the manuscript

throughout accordingly. Below is our point-by-point response to each of the comments.

General, fundamental comments

1. The analysis of what happened to the chorion after the depletion of DNMT1 and the increase and prolongation of *ftz-f1* expression can be further addressed by measuring the expression of chorion genes from different stages of choriogenesis (described in doi: 10.1186/1471-2164-10-206). For example, the authors could measure the expression of cathepsin-l (very early choriogenesis), yellow-g (early choriogenesis), citrus (early-mid choriogenesis), brownie (late choriogenesis), to see approximately when the process fails.

Response: Thanks for your constructive suggestion. We analyzed the expression of these five chorion genes (*cathepsin-L*, *yellow-g*, *citrus*, *brownie*, and *Fcp3c*) on both day 7 and day 9 PAE from our transcriptomic data (revised Fig. S4C), and qRT-PCR was performed to validate the expression of these five genes, at least four of these chorion genes were down regulated in the *dsDnmt1* group on day 9 PAE instead of day 7 PAE (revised Fig. 6D).

The corresponding changes in Figures and text are shown below:

“Moreover, the expression of five well-known chorion genes, *cathepsin-L*, *yellow-g*, *citrus*, *Fcp3c*, and *brownie*, in the transcriptome were explored. There was no significant expression difference for all of them on day 7 PAE, and three of them (*yellow-g*, *citrus*, and *Fcp3c*) were found significantly downregulated in the *dsDnmt1* group on day 9 PAE (Fig. S4C). The expression levels of them were validated by qRT-PCR, while *cathepsin-L* was found also downregulated significantly when detected by qRT-PCR (Fig. 6D).”

Fig.S4

Fig.6

In addition, the related references you suggested have already been inserted in the revised manuscript.

2. Analysis of transcriptomes. The effort made in the preparation and analysis of the transcriptomes deserves to be exploited more efficiently and specifically. The authors could go deeper into the DEGs, identifying which chorion genes are differentially expressed, and validating the transcriptomic values by qRT-PCR. This would help to disambiguate expressions like “leading to the failure to synthesize sufficient chorionic protein” (line 283), and have a general view of chorion-related genes affected by the experiment.

Response: Thanks again for your suggestion about the analysis of chorion genes. As we responded above, the chorion-related genes in the transcriptomes have been further exploited from the DEGs, and another qRT-PCR was performed to validate the expression of these chorion related genes. We have added these important data in revised manuscript (revised Fig. 6D and S4C).

3. Structure of the manuscript and Discussion. The Results section is complex, but the structure seems adequate, each section leading naturally to the next. In contrast, the Discussion looks poor and redundant. This section should be revised, in terms of discussing the results in relation to other contributions, not only in *Blattella* (there are a good number of contributions that can be compared with the results of Zhao et al.) but also in other insects. The Discussion could also address the debate on how DNA methylation affects gene expression, considering not only insects (doi: 10.1093/molbev/msw264), but animals in general (see, for example, doi: 10.1186/s13072-018-0205-1). The authors can also comment on possible future directions. For example, an interesting question that arises now is to figure out what determines the increase in DNMT-1 expression precisely during choriogenesis.

Response: Thanks for your constructive suggestion about the Discussion section. We have added all the information you mentioned above in the revised manuscript, and we did a thorough revision on the Discussion section.

The corresponding changes in Discussion are shown below:

“Moreover, in both vertebrates and insects, there’s still a general debate about the facilitative or preclusive effects of DNA methylation on gene expression^{42, 43}. In this study, a large number of both

upregulated genes and downregulated genes were observed after *Dnmt1* depletion, while the number of upregulated genes were much higher than downregulated genes (Fig. 2B), which indicates DNMT1-mediated DNA methylation mainly function as gene repressor in the German cockroach. Notably, an interesting question that arises now is how the expression of *Dnmt1* is precisely regulated during choriogenesis.”

General formal comments

4. Nomenclature of fushi tarazu factor 1. Please use the full name in the first mention. Then abbreviate it as *ftz-f1* (in italics, with lowercase initial, not capitalized as seen in most sections of the manuscript) when referring to the gene, and as FTZ-F1 when referring to the protein.

Response: Thanks for your kind suggestion. We have replaced these two words throughout the manuscript, figures, and legends.

5. Teleological expressions. I refer to expressions like “is a crucial epigenetic modification FOR regulating gene expression” (line 23-24), or “Ftz-f1 promoter undergoes 5mC modification TO prevent prolonged expression” (line 202), or “induces the expression of matrix metalloproteinase (Mmp) genes TO regulate the maturation and rupture of follicle cells” (lines 241-242) (capitals are mine). These expressions sound teleological, that is, they denote purpose, and there is no purpose in developmental and evolutionary biology processes, there is simply natural selection. It is easy to avoid the "FOR" or "TO" using alternative wording. For example: “is a crucial epigenetic modification INVOLVED IN regulating gene expression” (line 23-24), or “Ftz-f1 promoter undergoes 5mC modification THUS preventing prolonged expression”, or “induces the expression of matrix metalloproteinase (Mmp) genes THUS regulating the maturation and rupture of follicle cells”.

Response: Thank you for your suggestions. We have changed all these kinds of expressions according to your suggestion throughout the manuscript.

6. Background on Blattella studies. The authors can be interested in reading papers that are relevant to the work presented. For example, about vitellogenesis: DOI: 10.1111/j.1365-3032.1995.tb00801.x; DOI: 10.1016/s0965-1748(99)00058-2; DOI: 10.1111/j.1742-4658.2005.05066.x; DOI: 10.1016/0012-1606(86)90143-0. About choriogenesis: DOI: 10.1016/0040-8166(93)90019-h; DOI: 10.1186/1471-2164-10-206; DOI: 10.1016/j.ibmb.2010.11.001. About epigenetics and choriogenesis: DOI: 10.1016/j.yexcr.2013.07.006. About ftz-f1, choriogenesis, cell death: doi.org/10.1111/imb.12866; DOI: 10.1016/j.ydbio.2010.07.012; DOI: 10.1242/dev.190066. Etc.

Response: Thanks a lot for your kind suggestions. We have added many of these nice references in multiple sites of the revised manuscript.

Specific comments

7. The sentence “Reproduction is a fundamental process for all known organism, as each individual organism is the result of reproduction” (line 52-53) is trivial and unnecessary. The next one can perfectly serve as the initial sentence of the section.

Response: Thanks for your suggestion. We have removed this sentence.

8. Line 55. “production of yolky eggs”, read “production of yolky oocytes”. “laying of fertilized eggs”, read “laying of eggs”. “As the organs responsible for egg production (oogenesis)”, read “As the organs responsible for oogenesis”.

Response: Thanks. We have revised them according to your suggestions.

9. Line 63. “In most, if not all, insects, 20E regulates the latest stage of oogenesis, choriogenesis 16, 17”. A relevant work in *Blattella*: DOI: 10.1016/0040-8166(93)90019-h

Response: Thanks for your suggestion. We have added this reference in the revised manuscript as ref. 18.

10. Line 78. “Female reproduction and its regulation have been well studied in the German cockroach 23.”. Reference 23 by itself is not representative of the extensive work published on *Blattella* reproduction. See my comment 6 above for additional information.

Response: Thanks for your suggestion. We have added four more references you mentioned in comment#6 in the revised manuscript.

11. Line 110. “Abundant energids in the yolk mass were observed in the dsCK-treated eggs, which tended to concentrate at the ventral surface at 60 h and developed into early embryos at 84 h 24.”. Reference 24 does not describe this. This is described in DOI: 10.1016/j.ibmb.2010.04.006 and in DOI: 10.1016/j.isci.2018.05.017

Response: We apologize for the misuse of the references. We have changed the references according to your suggestion.

12. Line 164. “PAE. In accordance with the situations in other species, methylation modifications and changes were found at the cytosines of the CpG motif”. This has been reported specifically in *Blattella*: DOI: 10.1016/j.isci.2020.101778

Response: Thanks for your comment. We have added this reference at the end of this sentence.

13. Line 188. “the transcription factor gene *ftz* transcription factor 1 (*Ftz-f1*)”. See my comment 4, above.

Response: Corrected. Thanks again.

14. Line 199. “These findings suggest that FTZ-F1 acts not only as a transcription factor promoting steroidogenesis but also as a competence factor that works in conjunction with Tai to enhance EcR activity for 20E signalling 29, 30.”. This cannot be inferred from the empirical evidences reported.

Response: Thanks for your constructive suggestion. To avoid overstatement, we changed “acts” to “might act”.

15. Line 220 “The co-treatment of dsDnmt1 increased the depleted *Ftz-f1* expression caused by ds*Ftz-f1* treatment. In other words, the co-treatment of ds*Ftz-f1* attenuated the increased and prolonged *Ftz-f1* expression caused by dsDnmt1 treatment (Fig. 5A).” This is quite confusing. One (good and straightforward) sentence would be enough.

Response: Thanks for your suggestion. We have simplified these two sentences to single sentence: “The co-treatment of ds*ftz-f1* attenuated the ds*Dnmt1*-induced *ftz-f1* upregulation (Fig. 5A).”

16. Line 224. “Since dsFtz-f1 treatment can completely disrupt SPAWNING process, the co-treatment of dsDnmt1 increased the SPAWNING rate back...” (capitals are mine). Spawning does not exist in cockroaches.

Response: We apologize for this mistake, and we have changed it to “oviposition” in revised manuscript.

17. Line 228. “These genetic interaction”. Ambiguous. What genetic interactions are the authors referring to?

Response: We apologize for the confusing of this formulation, we have changed it to “The co-injection results of ds*Dnmt1* and ds*ftz-f1* further demonstrate that...”.

18. Line 249 to 254. This paragraph fits better in the Discussion section.

Response: Thanks for your suggestion, and we moved this paragraph to the Discussion section.

19. Line 266. “the 20E PURGE should be responsible for inducing Ftz-f1 expression” (capitals are mine). I presume the authors mean “20E pulse”.

Response: Corrected, sorry for this typo.

20. Line 283. “leading to the failure to synthesize sufficient chorionic protein”. There are well characterized chorion genes well characterized in *Blattella*, like citrus (early chorion), brownie (sponge like body) etc. (see my point 1 above) that could be measured. The detailed analysis of the transcriptomes could also reveal chorion proteins that are “insufficient” (see my point 2 above).

Response: Thanks for this suggestion about chorion genes again. As we responded above, these data have been added in the revised manuscript.

21. Line 317 and the following. Please indicate what is the unit of measurement in quantitative gene expression (I presume copies per copy of actins)

Response: Thanks for the remind, we have added the relative information in the legend and Materials and Methods section.

22. Line 433. *Blattella germanica* must be in italics.

Response: Corrected to italic *B. germanica*.

23. Reference 11. Instead of citing a book review, the authors might cite the book itself.

Response: Thanks for your suggestion, we have changed this reference to the book itself.

24. Figure 1A. “Throax”. Read “Thorax”.

Response: Corrected, sorry for this typo.

Reviewer #2 (Remarks to the Author):

Here, Zhao et al aim to unravel how CpG methylation regulates female reproduction in the cockroach *Blattella germanica*. In their manuscript, Zhao et al present two stories. First, they provide RNAseq of DNAmethylase (Dnmt1) knock-down and control ovaries at days 7 and 9 post adult emergence, as well as whole-genome bisulphite sequencing of Dnmt1 knockdown and control ovaries at day 9. Second, they provide a functional study of the candidate genes *Ftz-f1* and *Mmp1* in ovary development, including analysis of the methylation pattern of the two candidate genes. The first story would have been suitable for publication in Nature Communications if an unbiased approach in identifying pathways and candidate genes would have been taken. The second study is well performed, but may be more suitable for a specialized journal. The main problem with the manuscript is that these two stories are quite unrelated, or at least: the second story does not at all logically follow from the first, as I explain in the three paragraphs below, and conclude in the fourth paragraph.

Response: Thank you so much for the evaluation of our manuscript. We highly appreciate your critical comments on our work, which are very helpful for improving this paper. Sorry for the possible misleading in our manuscript organization. Your suggestions will certainly make the paper logically clearer, forming a fluent and complete story. Our logic is as follow: first, we discover that DNMT1-mediated 5mC maintenance is indispensable for choriogenesis and fertilization through repressing excessive 20E levels (Fig. 1 and 2). Second, we find that DNMT1-maintained 5mC levels at the *ftz-f1* promoter region repress *ftz-f1* expression that regulates proper steroidogenesis and 20E levels (Fig. 3-5). Last, we show how FTZ-F1 controls choriogenesis and fertilization (Fig. 5 and 6). Thus, our simple purpose of this manuscript is to figure out the detailed molecular mechanism how the DNMT1-mediated 5mC maintenance regulates female reproduction in the German cockroach, *Blattella germanica*, particularly at the transcriptional level. To address your concern, we drew a new flowchart (revised Fig. S3A), performed more experimental screening work from RNA-Seq data and thus focused on 20E (revised Fig.S2A-S2C), and performed more experimental screening work from WGBS data and thus focused on *ftz-f1* (revised Fig. 3E-3F and S3E). We hope our additional work and efforts could convince you with a clean story.

To start, it is a bit unclear what the main question of the paper is. In the abstract, the authors write: "DNA methylation...is a crucial epigenetic modification for regulating gene expression, but little is known about its regulatory function and the underlying mechanism in insects" (lines 23-25), and: "This study significantly advances the understanding of how DNA 5mC modification regulates...female reproduction in insects" (lines 35-36). In the introduction, the authors write: "However, there is limited knowledge regarding the mechanistic links between 5mC modification and insect development, particularly regarding transcriptional regulation" (lines 48-50), and: "...we identified how 5mC modification regulates key aspects of female reproduction" (lines 81-82). Although no single clear question is presented, the reader now gets a feeling of what the authors want.

Response: We apologize for the confusing. To make the logic clearer, in the revised version, we clearly stated our scientific question at the second sentence in the Abstract "Here, we pursue detailed molecular mechanism how the DNMT1-mediated 5mC maintenance regulates female reproduction in the German cockroach, *Blattella germanica*." Similar statement is also shown at the end of the

first paragraph in the Introduction. Meanwhile, at the ends of the Abstract and the third paragraph in the Introduction, we summarized the major discovery, in consistent with the title “5mC modification orchestrates choriogenesis and fertilization by preventing prolonged *ftz-f1* expression”.

The reader now expects things such as knockdown of methylating enzymes, bisulfite sequencing and RNAseq, exactly like the authors present, followed by an unbiased functional analysis of the most outstanding and promising pathways and candidate genes. The key problem is that the authors do not do the latter. When they analyse the RNAseq data in line 146, the authors state out of the blue: “Our analysis revealed that the insect hormone biosynthesis pathway was enriched (Fig. S2).” But when I look at Figure S2, “Insect hormone signaling” is only a minor category. Why do the authors choose this category for further study? There are many more and much higher over-represented pathways, such as ubiquitination, endocytosis, or wnt signaling. Why not going into these? The authors give no explanation, and this is the main problem of the manuscript. They go on in lines 146-148: “Further examination of the upregulated genes in this pathway identified phantom (Phm) and shadow (Sad), both of which are involved in steroidogenesis (Fig. 2C)”. Yes, they now specifically look at Phm and Sad and show mild upregulation. Much more interesting would be to show where Phm, Shd (or the later analyzed *Ftz-f1* and *Mmp1*) are in the volcano plot in Figure 2B. Do they belong to the genes that show the largest upregulation? I don’t think so. What are actually the most outstanding dots in Figure 2B? This would be a way to objectively identify functionally involved genes in an unbiased way.

Response: Thank you so much for your insightful comment. We completely agree that the main problem of the manuscript is why we directly go to 20E. Please let me explain:

At the beginning of this study, we even did not know the physiological function of DNMT1-maintained 5mC modification in the German cockroach. After a long trial, we found that 5mC modification regulates female reproduction, that is why we raised the scientific question “Here, we pursue detailed molecular mechanism how the DNMT1-mediated 5mC maintenance regulates female reproduction in the German cockroach, *Blattella germanica*.” After careful morphological observations, we then focused on choriogenesis (Fig. 1). The transcriptomic changes were consistent with the morphological changes observed during choriogenesis (Fig. 2A, 2B and 1I). In our original manuscript, we analyzed the RNA-Seq data and performed the KEGG enrichment analysis (just the count of genes in each term/category are considered in original Fig. S2) for the upregulated genes. Many of categories such as insect hormone biosynthesis, ubiquitination, endocytosis, or wnt signaling were enriched (original Fig. S2). As shown in the second paragraph of Introduction (In most, if not all, insects, 20E regulates the latest stage of oogenesis, choriogenesis^{16, 17, 18.}), 20E is the predominate regulator of choriogenesis in insects, then we went straightforward to 20E.

To avoid this bias, we performed the KEGG enrichment using more parameters (both the count of genes and the *p*-adjust were considered), the terms of ubiquitination, endocytosis, or wnt signaling no longer been enriched, while some categories about nutrition related pathways, and importantly, the insect hormone biosynthesis, were enriched (revised Fig. S2A). According to your suggestion to screen, we detected the effects of nutrition associated components on cockroach female reproduction. When injected glucose, insulin, and amino acids mixture *in vivo*, no obvious phenotypic changes similar to *dsDnmt1* treatment were found. Particularly, there was no significant decrease about the embryogenesis rate or oothecae hatching rate observed in the *dsDnmt1* group (revised Fig. S2B and S2C). Thus, we focused on the insect hormone biosynthesis pathway, since

phantom (*Phm*) and *shadow* (*Sad*), both of which are involved in steroidogenesis, were identified upregulated in ds*Dnmt1* group (Fig. 2C). In the following, we observed the 20E treatment phenocopied ds*Dnmt1* treatment at a great degree. Thus, we have enough confidence to explore the potential regulation of 5mC modification on 20E.

I have worked on insect hormone for 30 years and published over 150 papers related to insect hormones, including some significant publications in *Nature Ecology and Evolution*, *Nature Communications*, *PNAS*, *Autophagy*, *Molecular Biology and Evolution*, and *Development*. One major reason is we found that ds*Dnmt1* regulates 20E, that is why I decide to go deeper into this objective.

The same happens when the authors add the bisulfite sequencing data. Out of the blue, the authors write in lines 188-189: “Subsequently, we analyzed the hypomethylated and upregulated genes in top-left quadrant and identified the transcription factor gene *ftz* transcription factor 1 (*Ftz-f1*)”. Which of these dots in the upper left quadrant of Figure 3C actually is *Ftz*? Does it stand out? What genes do stand out? Again, the choice for further study seems biased. To make the problem complete, *Phm* and *Sad* do not show any differential methylation, and neither does the later studied *Mmp1* (Fig. S4). Hypomethylation of the main gene *Ftz* is later shown by targeted bisulfate sequencing PCR, but *Ftz* does not stand out in the original whole-genome approach.

Response: Thank you so much for your helpful suggestion. First of all, to make the logic clearer and let the audience understand easier, we drew a new flowchart regarding to the *ftz-f1* studies (revised Fig. S3A).

Once we focused on 20E signaling and neither *Phm* nor *Sad* show any differential methylation, we then searched for the genes whose expression might be controlled by 5mC modification and involved in regulating steroidogenesis. In one side, *ftz-f1* was screened out from the steroidogenesis associated transcription factor genes (*EcR*, *E75*, *E93*, and *ftz-f1*. DOI: 10.1242/dev.102020) (revised Fig. 3C), showing that DNMT1 might inhibit steroidogenesis through maintaining 5mC levels of the promoter regions of *ftz-f1*. In another side, all the hypomethylated and upregulated transcription factor (TF) genes in top-left quadrant were screened (revised Fig. 3E). Among them, the top six TF genes (>2-fold upregulation) were knocked down one by one (revised Fig. S3E), and the second-ranked upregulated *ftz-f1* was found significantly regulated the expression of *Phm* (revised Fig. 3F). Thus, we focused on *ftz-f1* and its function in female reproduction. In the following, we demonstrate that DNMT1-maintained 5mC levels in the *ftz-f1* promoter regions inhibit *ftz-f1* expression (revised Fig. S3F and Fig. 3). Notably, the co-treatment of ds*ftz-f1* partially rescued ds*Dnmt1*-induced follicle cell disorganization and sponge-like body malformation (Fig. 5D and 5E). Thus, we made a clear conclusion that ds*Dnmt1*-increased and prolonged *ftz-f1* expression disrupts choriogenesis and fertilization.

We agree with you other ds*Dnmt1*-regulated genes might be involved in female reproduction. As we emphasized in the manuscript, “exogenous 20E treatment only partially mimicked ds*Dnmt1*-induced phenotypic defects, which suggests more genes or factors are directly affected by ds*Dnmt1* treatment (Fig. 3D)” (Discussion section).

In conclusion, the authors provide a very good study on how *Ftz* and ecdysone are involved in female ovary development and oogenesis. They do show that methylation of *Ftz* is involved. This is a lot of work and highly interesting for a more specialized journal. However, the authors fail to

provide an unbiased analysis of targets of methylation, and are far from elucidating how methylation “precisely regulates insect reproduction”.

Response: As for the screening of 20E signaling and *ftz-f1*, we hope the efforts and explanations we made could convince you now. We highly appreciate your constructive comments and suggestions, which are very helpful for improving the manuscript with much less bias. I learned a lot from your deep scientific insights and critical suggestions. Thank you from the bottom of my heart.

Just like the Reviewer#1 said: “*This result is important not only in the context of the specific issue of oocyte chorionation, but also in the context of the general debate about the effects of DNA methylation and gene expression. This makes the manuscript interesting to an audience larger than that of insect molecular physiologists.*” We believe this revised manuscript will be interested by general audiences of Nature Communications.

Experimental issues:

-In figure 1e, the authors show that fertilization does not take place upon *Dnmt1* RNAi. This is of course crucial. Otherwise, it could be that the embryos start to develop, but that failure of embryonic development is due to *Dnmt1* knockdown in the embryo, i.e. aberrant methylation during embryonic development. On how many eggs is this conclusion based? Same question for Figure 1F: how many eggs have been analysed? And for Figure 1I: how many follicles have been analyzed? N should be crystal clear and indicated in the figure legend.

Response: Thanks a lot for your comments. Actually, the conclusion “fertilization does not take place upon *Dnmt1* RNAi” is made according to all the whole data in Fig.1. As we mentioned in the manuscript about figure 1E, we just found the oothecae were atrophied and wizened. To search the reason for *dsDnmt1*-induced phenotype, we did a series of backward tests and found the failure embryonic development (Fig. 1F), failure sperm pronucleus staining (Fig. 1G), abnormal sponge-like bodies (Fig. 1H), and follicle cells cytoskeletal disorganization (Fig. 1G). These results together tell us the *dsDnmt1*-induced failure of fertilization is mainly resulted from the non-functional of sponge-like body (an atretic structure appearing at the anterior pole of the basal oocyte through which sperms enter maturing egg for fertilization) instead of aberrant methylation during embryonic development. We hope our explanation will meet your satisfaction.

Thanks again for your suggestion about the sample numbers. We have added all these kinds of details in the revised figure legends.

-Lines 154-155: “while the mating and ootheca-laying rates of the insects were unaffected” Where are the data that show this?

Response: We apologize for this unclear description. We think this result is not tightly involved in this story, we have removed this sentence from the revised manuscript.

-Lines 156-157: “most of the sponge-like bodies in the 20E-treated animals were malformed” How many sponge-like bodies have been investigated, and how many were malformed?

Response: Thanks again for your insightful suggestion about the sample numbers. We have added this detail information in the revised manuscript.

-Figure 3B: Why do we see an increase in hypermethylation upon *Dnmt1* RNAi? Isn't this

unexpected, as one would expect a decrease in methylation when a methylation enzyme is knocked down?

Response: Thanks for your important suggestion. We agree with you that we expect a decrease in methylation under *dsDnmt1* treatment, and there indeed 205842 (99.97%) hypo-DMRs were detected. As for the 57 (0.03%) hyper-DMRs, we think this low probability rate (<0.05%) might be (1) the truth in the ovaries after *dsDnmt1* treatment, (2) the acceptable error range of WGBS method, or (3) the effect of other methylation enzyme like DNMT3.

-Line 224: I am not a native speaker, but “spawning” suggest to me something organisms can only do in an aquatic environment. What do the authors mean? How is the “spawning rate” in Figure 5C determined?

Response: Thanks for your critical suggestion, we have changed this word to “oviposition” in revised manuscript.

Minor comments:

n; the result is 5mC. Why not simply calling it CpG methylation, as many other papers do? The authors nicely prove that it is indeed CpG methylation, and not any other sites such as CHG or CHH (line 166).

Response: The reason for the usage of 5mC (DNA methylation at the fifth position of cytosine) here is try to distinguish this modification with other less known methylation types, such as 4mC and others.

-Maybe the authors could mention in the legend of Figure 1a what the staining for “input” is.

Response: Thanks for your important suggestion, we have added this information in the revised legend and Methods.

-Fig1a': does oviposition take place after day 8?

Response: Yes. There are 8 days in the first reproductive period of German cockroach in our lab rearing condition. That's why we selected these 8 timepoint samples to detect the modification levels. To make this description much clearer, we modified this sentence to “the 5mC temporal pattern in the ovaries during the whole first reproductive period (8 days) was evaluated” in revised manuscript.

-Fig. 1e: what do the small pictures show? Please explain in the legend.

Response: Thanks for your helpful suggestion. The small pictures in figure 1E are abnormal phenotypes (atrophied and wizened) of oothecae in *dsDnmt1* group compared with *dsCK* control group. We have explained this in the revised legend according to your suggestion.

-Lines 110-115: could the authors provide references in the text to the exact sub-panel of Figure 1F that show what the authors describe. Maybe some arrows or a better description in the legend would help the reader.

Response: Thanks for this very important suggestion. We have added the arrows in the revised figure 1F to emphasize the details during embryo development. And related information has been added in the revised legend.

-Figure S1F: “the main component Vg protein in the oocytes was marked”. What do the authors mean? That the thickest band is Vg? How do the authors know?

Response: We apologize for the missing of the marker. We have added it in the revised figure S1F and the legend.

-Lines 142-143: “These transcriptomic changes were consistent with the morphological changes observed at both time points” In what sense? That there is no effect on day 7, but a clear effect on day 9?

Response: We apologize for the unclear description. Your understanding is absolutely correct, and we have modified this sentence to “These transcriptomic changes were consistent with the morphological changes observed at both time points, with insignificant effects on day 7 but significant effects on day 9 (Fig. 1I)” in revised manuscript.

-Figure S3D: What does Dmel mean? Demethylated? As I also see Bger (which probably means *Blatella germanica*), it could also be *Drosophila melanogaster*? Please explain the legend. What is actually analysed and shown here? Are only transcription factors taken into account? Why?

Response: We apologize for the unclear description of Bger and Dmel. Your understanding of both of them are correct. The intersecting genes between 413 upregulated DEGs and 259 trusted transcription factors (TF) in *Drosophila melanogaster* (Dmel) were analyzed. We have modified this part in revised Fig.3E and legend. “Subsequently, we analyzed the 413 hypomethylated and upregulated genes in top-left quadrant, eight transcription factor (TF) genes, *huckebein (hkb)*, *ftz-f1*, *Mi-2*, *Hormone receptor-like in 39 (Hr39)*, *bric a brac 2 (bab2)*, *Cyclic-AMP response element binding protein A (CrebA)*, *nejire (nej)*, and *deformed wings (dwg)*, which might regulate the expression of steroidogenesis genes were screened (Fig. 3E). Next, the top six TF genes (>2-fold upregulation) were knocked down one by one (Fig. S3E), and only *dsftz-f1* treatment significantly decreased the expression of *Phm* (Fig. 3F)”. The reason for the screening of TF genes is that these genes with high possibility to regulate the expression of steroidogenesis genes.

-Line 198: “*Phm*, *Ftz-f1*, *EcR*, or *Taiman*” Why were these genes chosen? What kind of gene is *Taiman*? *Taiman* comes out of the blue and could use some introduction.

Response: Thanks for your kind suggestion. *Phm* (steroidogenesis gene), *EcR* (ecdysone receptor), *ftz-f1*, or *Taiman* (the co-activator of *EcR*) are key candidate genes involved in steroidogenesis and 20E signaling. *Phm* and *EcR* are regulated by *ftz-f1* (revised Fig. 3F and Fig. S3G). We chose these four genes to detect whether their knockdown treatments will decrease the 20E level. We found “knocking down each of *Phm*, *ftz-f1*, *EcR*, or *Taiman* (the co-activator of *EcR*) significantly decreased 20E levels (revised Fig. 3G and Fig. S3H-S3J). These findings suggest that FTZ-F1 acts not only as a transcription factor promoting steroidogenesis but also might as a competence factor that works in conjunction with Tai to enhance *EcR* activity for 20E signaling”. According to your suggestion, the introductive information for these genes have been added in their first mention.

-Line 205: “the comprehensive 5mC level around the *Ftz-f1* gene in the genome was checked”. How? I guess looking at the whole-genome bisulfite sequencing data?

Response: Thanks for your important suggestion. As mentioned in revised manuscript “the

comprehensive 5mC level around the *ftz-fl* in the genome was checked from WGBS data”.

-What is the size of the Ftz gene/locus? Figure 4B suggests that the gene is around 8kb, i.e. an order of magnitude that can be expressed as 10³ bp. The x-axis and gene model on panel 4A, however, suggest an order of magnitude of 10⁵ bp.

Response: Thanks for your kind suggestion. The length of *ftz-fl* ORF is 1809 bp (10.1111/imb.12866)(10.1038/s41559-017-0459-1), and the gene body length (including all exons and introns) is about 4.48x10⁵ bp (Figure 4A). The 8 kb in Figure 4B indicates the upstream promoter region of *ftz-fl* gene, which including the classical up 2k promoter and another far away region. We have added more detail information in the revised manuscript and legend.

-Figure 4C: These 6 amplicons are different from the 6 amplicons in Figure 4B. Where exactly are these amplicons of panel 4C located? Maybe the authors can draw something similar like the schematic drawing in 4B. And how do the authors define “the promotor” in line 209? Simply the 2kb upstream sequence?

Response: We apologize for this unclear description. The 6 amplicons used in Figure 4C are same to that in Figure 4B. The corresponding changes in revised text are shown below:

“We also evaluated the temporal pattern of the 5mC level in the promoter of *ftz-fl* during the first gonadal cycle using NGS-BSP. The same 6 amplicons were detected again, and there was an evident decreasing trend in amplicons 3 and 5 from day 1-3 PAE to day 4-6 PAE that was followed by an increase until oviposition (Fig. 4C)”.

Lines 235-237: could the authors help the reader a bit with what should be seen in the electron micrographs? (maybe some arrow: where do we see dissociation?)

Response: Thanks for your helpful suggestion. We have added the asterisks in revised Fig. 6A and related legend.

-Line 237: why are the authors suddenly going to look at caspase3? For the general audience of Nature Communications, the function of caspase3 should be introduced and a rationale to study this gene should be given.

Response: Thanks for your helpful suggestion. Caspase-3 is one of the cell death marker related to anoikis, which process will lead to cell detachment. We have modified these information as “Moreover, the activity of cell death marker Caspase-3 was dramatically increased in follicle cells of *dsDnmt1*-treated insects, and *dsDnmt1*-induced Caspase-3 activity was blocked by the Pan Caspase inhibitor Z-VAD-FMK (Fig. 6B). These results suggesting that the *dsDnmt1*-treated follicle cells were likely undergoing anoikis, a cell death process related to detachment” in revised manuscript.

-Line 446: Who of the authors is E.D.?

Response: Thanks for your kind remind. We have removed this irrelevant information from the revised manuscript.

Reviewer #3 (Remarks to the Author):

I think the present investigation clarified the role of 5mC modification on chorionation. It also confirmed the inhibitory mechanism by regulating Ftz-F1 expression. The methodology is sound and the results are convincing. I therefore support publication of this manuscript after minor English correction. I am curious how *dnmt1* will be turned on or regulated upon starvation that induces oosorption. Comments of this question may be desirable.

Response: Thanks. We are very grateful for your high evaluation for our manuscript. We have double checked the language throughout the manuscript.

About the effect of starvation on *Dnmt1*, we detected the gene expression of *Dnmt1* in ovary by using qRT-PCR. There was a significant upregulation of *Dnmt1* under starvation (see the figure below). However, we think this result is not tightly involved in this study, we might do more study in future as another story.

** Please also note that Reviewer 3 has included a marked-up manuscript, which is attached to this message.

Response: Thanks a lot for your reminding. We are appreciated for the reviewer's careful and throughout reading of our manuscript, all the comments in the PDF file has been modified in the revised manuscript.

REVIEWERS' COMMENTS

Reviewer #1 (Remarks to the Author):

I see that the authors have addressed all my comments, and the new version of the manuscript is very satisfactory. I would only add some minor correction suggestions referring to aspects of form.

1. Line 64. "larval-/nymphal-specific", I'd say "larval/nymphal-specific"
 2. Line 73. "Oncopeltus fasciatus 7, 20, 21, 22, 23". The reference "9", previously cited, should be added here to this list to cover the case of *B. germanica*.
 3. Line 150. "since phantom (Phm) and shadow (Sad)". Read "sad" and "pham", not capitalized. Please apply the same correction in other parts of manuscript For example, lines 179, 181, 202, 205...).
 4. Lines 164-165. "In accordance with the situations in other species, methylation modifications were found predominantly at the cytosines of the CpG dinucleotide 9". Reference 9 refers to *B. germanica*. So, "In accordance with the situations in other specie" should be replaced by "In accordance with previous studies in *B. germanica*".
 5. Line 206. "Taiman (Tai, the co-activator of EcR)". Please read "taiman (tai, the co-receptor of ecdysone together with EcR)" (genes in italics).
 6. Line 250. "*D. melanogaster*". Please read "*Drosophila melanogaster*" (in italics), as it is the first mention of the species.
 7. Lines 306-308. "Moreover, FTZ-F1 has been found controlling the degeneration and death of the prothoracic gland during the nymphal–adult transition in *B. germanica* 47". Ref 47 is OK, but the actual mechanism was complementarily elucidated later (see doi:10.1242/dev.190066). Indeed, towards the end of the last nymphal instar of *B. germanica*, FTZ-F1 stimulates the expression of E93, which is the effector of PG death. So, both, FTZ-F1 (ref 47) and E93 (ref doi:10.1242/dev.190066) are important for PG degeneration.
 8. Line 538. The complete title of the book is "Insect metamorphosis. From natural history to regulation of development and evolution".
- Xavier Belles

Reviewer #2 (Remarks to the Author):

The authors did an outstanding job addressing my criticism. The logic and flow of the paper improved substantially. Whether this paper fits the scope and ambition of Nature Communications is of course up to the editor, but I have no objections against the science and argumentation in the manuscript anymore.

Although I am also not a native speaker, I think that the English/style could still use some improvement throughout the manuscript. (e.g. lines 53 "excellent fecundity"; line 253 "found been downregulated"; lines 283-284 "which indicates...methylation mainly function as gene repressor...", and it is a bit unclear to what "it" refers in line 50).

Reviewer #3 (Remarks to the Author):

The text became clearer in the revised ms, especially in discussion. The study depicted that the dnmt is involved in epigenetic modification of reproduction in the german cockroach via Fts-f1. However, neuroendocrine regulation in reproduction is interactive within constituent factors such as JH and peptides and diverse among insects especially in the role of 20E. Discussion must examine the ubiquitous occurrence of this type of regulation among other species. Also, epigenetic system must be involved in different physiological adaptations but current focus is not focused sharp. I recommended the authors should check the effect of food deprivation and they have data. I think it is kinder to the reader to show this first. This should be the main line rather than 20E. It is an internal gadgetry of the black box but starvation is an important input signal.

Response to reviewers
5mC modification orchestrates choriogenesis and fertilization by
preventing prolonged *ftz-f1* expression
(NCOMMS-23-31301B)

We thank the reviewers' recognition for our efforts to improve the manuscript. We have fully revised our manuscript again and have addressed all the points raised by the reviewers, and we think the resulting manuscript is much improved. Specific revisions and point-by-point responses are provided below.

REVIEWER COMMENTS

Reviewer #1 (Remarks to the Author):

I see that the authors have addressed all my comments, and the new version of the manuscript is very satisfactory. I would only add some minor correction suggestions referring to aspects of form.

Response: Thank you so much for providing us with valuable, constructive suggestions to help us improve the manuscript. We are pleased that our efforts and additional work have convinced you of the high-quality nature of our story.

1. Line 64. "larval-/nymphal-specific", I'd say "larval/nymphal-specific"

Response: Corrected.

2. Line 73. "Oncopeltus fasciatus 7, 20, 21, 22, 23". The reference "9", previously cited, should be added here to this list to cover the case of *B. germanica*.

Response: Thanks for your suggestion, we have added the reference 9.

3. Line 150. "since phantom (Phm) and shadow (Sad)". Read "sad" and "pham", not capitalized. Please apply the same correction in other parts of manuscript For example, lines 179, 181, 202, 205...).

Response: Corrected.

4. Lines 164-165. "In accordance with the situations in other species, methylation modifications were found predominantly at the cytosines of the CpG dinucleotide 9". Reference 9 refers to *B. germanica*. So, "In accordance with the situations in other specie" should be replaced by "In accordance with previous studies in *B. germanica*".

Response: We have revised this sentence according to your suggestion.

5. Line 206. "Taiman (Tai, the co-activator of EcR)". Please read "taiman (tai, the co-receptor of ecdysone together with EcR)" (genes in italics).

Response: Corrected.

6. Line 250. "D. melanogaster". Please read "Drosophila melanogaster" (in italics), as it is the first mention of the species.

Response: Corrected.

7. Lines 306-308. “Moreover, FTZ-F1 has been found controlling the degeneration and death of the prothoracic gland during the nymphal–adult transition in *B. germanica* 47”. Ref 47 is OK, but the actual mechanism was complementarily elucidated later (see doi:10.1242/dev.190066). Indeed, towards the end of the last nymphal instar of *B. germanica*, FTZ-F1 stimulates the expression of *E93*, which is the effector of PG death. So, both, FTZ-F1 (ref 47) and *E93* (ref doi:10.1242/dev.190066) are important for PG degeneration.

Response: Thanks for your kind suggestion, we have rephrased this sentence as “Moreover, FTZ-F1 has been found controlling the degeneration and death of the prothoracic gland by stimulating the expression of *E93* during the nymphal–adult transition in *B. germanica*” in the revised manuscript, and this new reference has been added as well.

8. Line 538. The complete title of the book is “Insect metamorphosis. From natural history to regulation of development and evolution”.

Response: Sorry for the mistake, we have changed the title according to your suggestion.

Reviewer #2 (Remarks to the Author):

The authors did an outstanding job addressing my criticism. The logic and flow of the paper improved substantially. Whether this paper fits the scope and ambition of Nature Communications is of course up to the editor, but I have no objections against the science and argumentation in the manuscript anymore.

Response: Thank you so much for providing us with valuable, constructive suggestions to help us improve the manuscript. We are pleased that our efforts and additional work have convinced you with a clean story.

Although I am also not a native speaker, I think that the English/style could still use some improvement throughout the manuscript. (e.g. lines 53 “excellent fecundity”; line 253 “found been downregulated”; lines 283-284 “which indicates...methylation mainly function as gene repressor...”, and it is a bit unclear to what “it” refers in line 50).

Response: Thanks for your suggestion, we have double checked the whole manuscript again to improve the English.

Reviewer #3 (Remarks to the Author):

The text became clearer in the revised ms, especially in discussion. The study depicted that the dnmt is involved in epigenetic modification of reproduction in the german cockroach via Fts-f1. However, neuroendocrine regulation in reproduction is interactive within constituent factors such as JH and peptides and diverse among insects especially in the role of 20E. Discussion must examine the ubiquitous occurrence of this type of regulation among other species. Also, epigenetic system must be involved in different physiological adaptations but current focus is not focused sharp. I recommended the authors should check the effect of food deprivation and they have data. I think it is kinder to the reader to show this first. This should be the main line rather than 20E. It is an internal gagetry of the black box but starvation is an important input signal.

Response: Thank you so much for your suggestion. We agree with you that the reproduction in insects is regulated by multiple factors (e.g. JH, peptides, and nutrition), we have added this information in revised manuscript. And we agree with you that starvation might be the main factor in regulating the reproduction insects, actually we have demonstrated that nutrition play key roles on previtellogenesis through regulating the biosynthesis of JH (DOI: 10.1242/dev.188805). Even so, we have added this data as Fig.S5 in revised manuscript.